# Cytolytic circumsporozoite-specific memory CD4⁺ T cell clones are expanded during *Plasmodium falciparum* infection

Raquel Furtado [1,12,14], Mahinder Paul [1,14], Jinghang Zhang [1], Joowhan Sung [2,13], Paul Karell[1], Ryung S. Kim[3], Sophie Caillat-Zucman [4], Li Liang[5], Philip Felgner [5], Andy Bauleni[6], Syze Gama[7], Andrea Buchwald[8], Terrie Taylor[7,9], Karl Seydel[7,9], Miriam Laufer[8], Fabien Delahaye [10,11], Johanna P. Daily[1,2] ✉ & Grégoire Lauvau [1] ✉

Clinical immunity against *Plasmodium falciparum* infection develops in residents of malaria endemic regions, manifesting in reduced clinical symptoms during infection and in protection against severe disease but the mechanisms are not fully understood. Here, we compare the cellular and humoral immune response of clinically immune (0-1 episode over 18 months) and susceptible (at least 3 episodes) during a mild episode of *Pf* malaria infection in a malaria endemic region of Malawi, by analysing peripheral blood samples using high dimensional mass cytometry (CyTOF), spectral flow cytometry and single-cell transcriptomic analyses. In the clinically immune, we find increased proportions of circulating follicular helper T cells and classical monocytes, while the humoral immune response shows characteristic age-related differences in the protected. Presence of memory CD4⁺ T cell clones with a strong cytolytic ZEB2⁺ T helper 1 effector signature, sharing identical T cell receptor clonotypes and recognizing the *Pf*-derived circumsporozoite protein (CSP) antigen are found in the blood of the *Pf*-infected participants gaining protection. Moreover, in clinically protected participants, ZEB2⁺ memory CD4⁺ T cells express lower level of inhibitory and chemotactic receptors. We thus propose that clonally expanded ZEB2⁺ CSP-specific cytolytic memory CD4⁺ Th1 cells may contribute to clinical immunity against the sporozoite and liver-stage *Pf* malaria.

*Plasmodium falciparum (P. falciparum, Pf)* infection remains highly prevalent with an estimated 241 million cases of malaria worldwide and an estimated 627,000 deaths in 2019[1]. Malaria infection is also associated with significant morbidity including decreased cognitive function and anemia in school-aged children[2–4]. The *P. falciparum* parasite evades host immunity by preventing optimal T and B cell responses, and by subverting effective vaccination through a variety of mechanisms[5–7]. These include generating immense antigenic diversity, also through distinct stages of development[8], promoting inflammation[9–11], and inducing multiple immune suppressive mechanisms[12–16]. Nevertheless, residents of highly malaria-endemic regions with high transmission develop clinical immunity, a host state that protects them from severe illness and death despite infection. Patients with clinical immunity have mild to no clinical symptoms and have low parasite burdens that are often submicroscopic during infection[17–19]. Protection from severe disease and the lack of symptoms

during infection appears to be long lasting if residents remain in endemic regions[20,21]. The only WHO-recommended malaria vaccine, RTS,S/AS01, modestly reduces the number of clinical malaria episodes by 46% and severe malaria by 36%[22]. In addition, the efficacy of RTS,S/AS01 wanes over time, with only 2.5% efficacy four years after vaccination, with some vaccines having a rebound in the number of clinical malaria episodes during the fifth year[23,24]. Thus, a more effective and long-lasting vaccine is still needed. Studies of naturally acquired clinical immunity may shed light on protective immune mechanisms that could be leveraged in the development of a more effective vaccine. How natural protection is achieved remains poorly understood and could serve as a stepping stone to uncover immune mechanisms relevant to host immunity against *P. falciparum*.

Both innate and adaptive responses have been associated with host protection against *P. falciparum* infection and clinical disease, and long-term immunological memory relies on parasite-specific CD4[+] T, CD8[+] T, and B lymphocytes. In individuals vaccinated with irradiated sporozoites and subsequently challenged with homologous sporozoite parasites in human and animal studies, polyfunctional memory CD8[+] and possibly CD4[+] T cells help mediate robust liver stage protection[25–31]. The liver stage of infection represents a *Pf* parasite population bottleneck, where there is limited parasite genetic variation, and effective protection could be achieved. It is known that IFNγ and cytolysis of infected hepatocytes are involved, however, the exact effector mechanisms that are needed and the sequences of events leading to host protection are still being debated. Whether any of these mechanisms are implicated in naturally acquired immunity also remains to be determined.

Once merozoites are released from the liver to the blood circulation, which occurs within ~7 days in humans, parasite-specific antibodies (Ab), most notably IgG[32–35] but also IgM[36,37], represent essential mediators of naturally acquired immunity. The production of parasite-specific protective Ab is a lengthy and complex process that may take years of endemic exposure to malaria, likely as a result of multiple negative regulatory mechanisms[38]. Serum Ab from clinically protected patients react against a higher number of parasite antigens (Ag) and have increased titers compared to sera from susceptible individuals[38,39], but which parasite Ag need to be targeted and whether specific functional features of targeting Ab are necessary to achieve clinical immunity is still unknown. Parasite-specific CD4[+] T cells, especially follicular helper CD4[+] T (T$_{FH}$) cells, are required to produce protective Ab by parasite-specific B cells. However, the development of a CD4[+] T$_{FH}$ cell response that promotes effective and long-lasting parasite-specific protective Ab results from multiple competing mechanisms orchestrating CD4[+] T cell differentiation during infection. Malaria infections, both in humans and in mouse models, drive robust T helper 1 (T$_H$1) responses as a result of a highly inflammatory environment, in which CD4[+] T cells upregulate the T$_H$1 master transcriptional regulator T-bet, produce IFNγ, and have cytolytic potential[40–42], all of which contribute to protective antimalarial responses[43]. These T cells also express high levels of cell-surface CXCR3 and have poor T$_{FH}$ cell functional characteristics[9,44,45]. However, as the inflammatory response decreases in individuals developing naturally acquired immunity, less T$_H$1 polarized CD4[+] T cells develop and differentiate into CXCR5[+]PD-1[+]CXCR3[-] CD4[+] T cells which exhibit the functional phenotype of T$_{FH}$ cells and the ability to provide strong help to parasite-specific B cells. Highly differentiated IFNγ[+] CD4[+] T cells can also co-secrete the immunoregulatory cytokine IL-10, dampening the immunopathology associated with the excessive inflammatory responses driven by the parasite during acute infection[16,46,47]. Thus, CD4[+] T$_H$ cells are essential for effective protection against malaria, yet exactly how these subsets are related to each other during malaria infection, how they develop during acquisition of natural immunity and contribute to protection, is not

clear. The functional features that these cells need to acquire to provide an effective host antimalarial response are mostly unknown.

In the current work, we undertake a comprehensive, unbiased systems immunology approach to define humoral and cellular immune signatures during symptomatic *P. falciparum* infection in residents of a high transmission malaria endemic area in Malawi. Participants are classified as clinically susceptible or clinically immune based on the number of clinical malaria episodes per participant over an 18-month longitudinal study. Using these clinical classifications, we then characterize their *P. falciparum*-specific Ab response and conduct high dimensional cytometry by time of flight (CyTOF) and spectral flow cytometry phenotyping and single-cell transcriptomic analysis of their memory CD4[+] T cells to identify signatures associated with clinical immunity against malaria. Our results show that older participants with clinical immunity have greater parasite-specific Ab breadth, titers and merozoite opsonization capacity compared to younger protected and susceptible participants. Importantly, we reveal in malaria-infected participants the existence of a population of clonally expanded memory CD4[+] T cells that share identical TCR clonotypes and express a robust cytolytic signature and the ZEB2 master regulator of terminally differentiated effector cells. ZEB2[+] memory CD4[+] T cells react to the pre-erythrocytic stage circumsporozoite (CSP) *P. falciparum*-derived Ag and could play a key role in clinical immunity.

## Results

### Longitudinal study of *P. falciparum* infected patients in Chikwawa District, Malawi

To identify immune signatures associated with clinical immunity against malaria, we conducted an 18-month longitudinal study of pediatric and adult patients residing in a rural region of southern Malawi in the Chikwawa District. This is a *P. falciparum* malaria endemic area with an inoculation rate of 183 infective bites/person per year, with year-round and high transmission and where older age is associated with clinical immunity[48]. We enrolled children, adolescents and adults, and over-sampled younger participants to provide a sufficient number of participants who displayed variation in clinical immunity. One hundred and twenty patients, who were otherwise healthy with no chronic conditions were enrolled during an episode of symptomatic, mild malaria infection, which was confirmed by microscopy of the blood smear. Participants were then evaluated monthly and at interim study clinic visits during illness to monitor for malaria infection, record temperature and collect blood samples over the course of the study. Clinical reinfection was defined as symptoms consistent with malaria and >2500 parasites/μl of blood seen on microscopy. Ninety-seven participants completed the study at 18 months of follow up (Fig. 1A and Supplementary Fig. 1A). Two protected participants had sickle cell trait and were excluded from further analysis as sickle cell confers a unique protective mechanism against malaria infection[49].

### Selection of participants who vary in clinical immunity for in depth immune analyses

The age at which clinical immunity develops within a population is based on transmission intensity and other local population level effects and thus needs to be defined in each malaria endemic region[50,51]. To classify Chikwawa participants as either susceptible or clinically immune we enumerated the number of recurrent clinical malaria episodes over the 18-month longitudinal study. An infection was counted if the malaria genotype determined by sequence analysis was distinct from the prior infection or occurred at least two weeks after a prior infection in each participant. We then defined clinically susceptible participants as those with the greatest number of clinical reinfections (upper tertile, ≥3) and protected ones as those with the lowest number of clinical reinfections (lower tertile, ≤1) (Fig. 1B). As expected clinically susceptible participants were younger in age

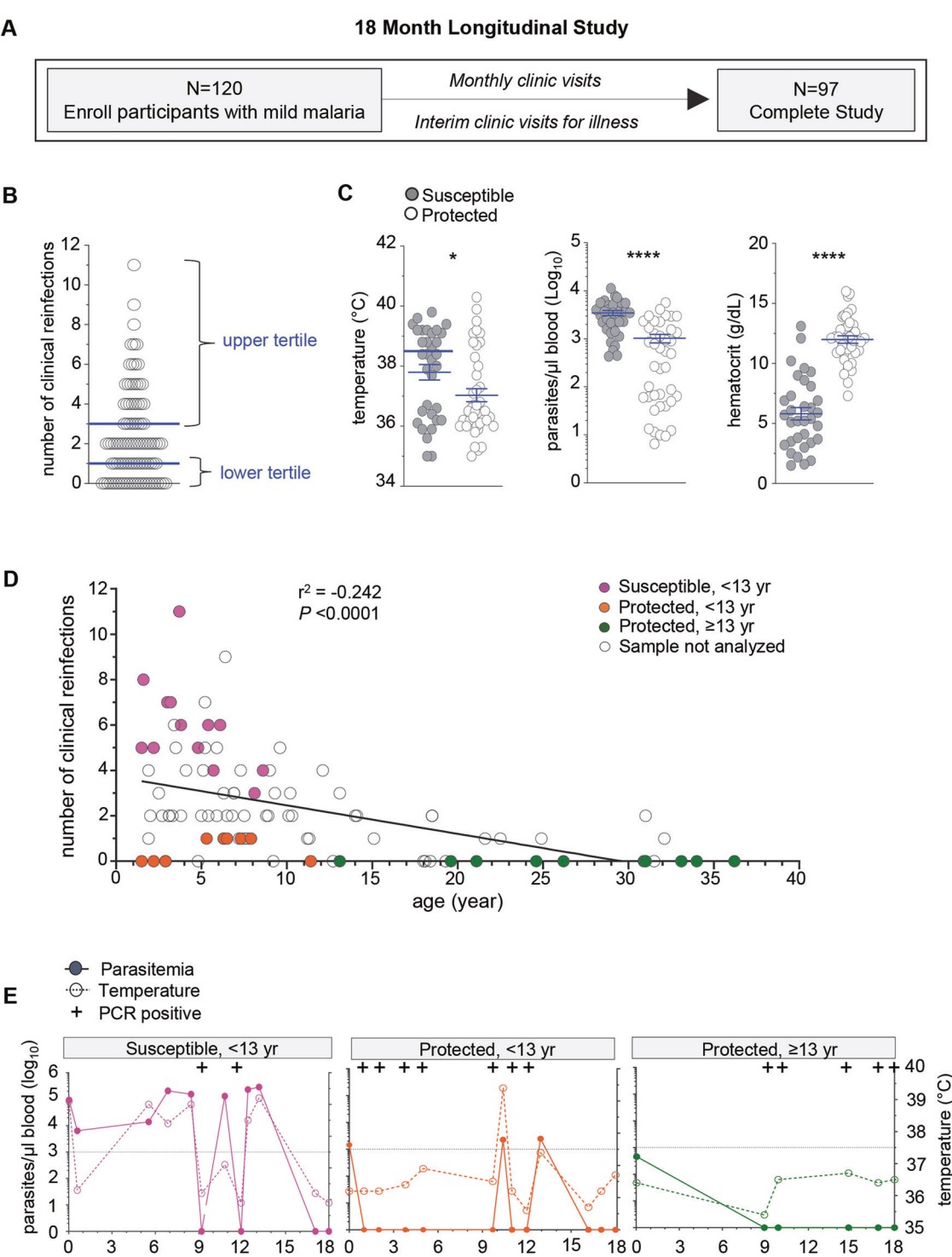

**Fig. 1 | Identification of participants with varied clinical immunity against malaria in a longitudinal study of an endemic region in Malawi. A** Overview of the 18-month longitudinal study of mild malaria participants with total enrollment number, clinical visits, and retention number at the end of the study. **B** Number of clinical reinfection episodes (symptoms, >2500 parasites/μl blood, with a prior episode at least 2 weeks apart) of each participant that completed the study. Upper tertile (67%) (≥3 clinical reinfections, susceptible, $n = 33$) and lower tertile (33%) (≤1 clinical reinfection, protected, $n = 41$) of participants are indicated. **C** Clinical parameters of temperature (°C), hematocrit (g/dL) and parasitemia (parasites/μl blood) of each protected (≤1 clinical reinfection, protected, $n = 41$) and susceptible participants (≥3 clinical reinfections, susceptible, $n = 33$) at the enrollment visit.

**D** Number of total clinical reinfection episodes (y-axis, >2500 parasites/μl blood, at least 2 weeks apart) and participants' age at time of enrollment (x-axis). Data was stratified by age, with susceptible participants (<13 yr, ≥3 clinical reinfections), protected participants (<13 yr, ≤1 clinical reinfection) and protected participants (≥13 yr, ≤1 clinical reinfection) are indicated. Pearson correlation coefficient and statistical significance are indicated. **E** Parasitemia (left axis), temperature (°C, right axis) and *Pf* PCR status indicated for each clinical visit of one representative susceptible, one protected age-matched and one protected adult participants over the 18-month longitudinal study. Dotted line is at 37.5 °C. Statistics: Two-sided Student's *t* test was conducted between indicated groups, *$p < 0.05$, **$p < 0.01$, ***$p < 0.001$, ****$p < 0.0001$. Data are presented as means ± SEM (**C**).

compared to protected individuals (Supplementary Fig. 1B). They had higher body temperature, greater blood parasite loads and lower hematocrits than the protected participants at the enrollment malaria episode (Fig. 1C). To determine if the greater number of clinical infections in younger participants compared to protected individuals was not due to higher exposure to malaria, we measured the total number of afebrile (temperature <37.5 °C), low blood parasite load (<2500 parasites/μl) and asymptomatic reinfections and the number of microscopy negative/PCR positive reinfections over the course of the study and found no differences between the groups in either scenario (Supplementary Fig. 1C). This suggested that the age-related differences in the number of clinical reinfections was not simply secondary to differences in *P. falciparum* infections, and thus likely related to immune status.

To conduct a subanalysis of the immune response among age-matched participants defined by their clinical immunity status, we age-stratified the clinically protected participants (lower tertile, ≤1, Fig. 1B) into two groups, e.g., younger than 13 years or 13 years or older (Supplementary Fig. 1D). In the less than 13 years of age group more males were found in the protected compared to the susceptible group (upper tertile, ≥3, Fig. 1B). Within these groups, we selected a set of participants for further analyses (filled symbols, Fig. 1D). Plots of representative individuals for all three groups reporting all infections (including microscopic and submicroscopic/PCR⁺), parasitemia and temperature over 18 months of follow up are provided (Fig. 1E and Supplementary Fig. 1E). In summary, we established a well-characterized cohort residing in a high transmission *P. falciparum* endemic area that demonstrated variation in clinical immunity. This enabled us to carry on deeper analyses including in depth immunophenotyping of their peripheral blood mononuclear cells (PBMC), analysis of their plasma Ab and single-cell transcriptomic to investigate host immune mechanisms associated with protection against symptomatic re-infections.

### *P. falciparum*-specific humoral responses in malaria immune compared to clinically susceptible participants

Ab recognizing malaria Ag play an important role against disease and are associated with clinical immunity[30,32,48,52,53]. To examine the link between anti-malarial Ab responses and clinical immunity we characterized the Ab response from participants' enrollment plasma samples during clinical malaria using a *P. falciparum* protein array displaying ~1130 predicted and known *P. falciparum* proteins[39,54,55] (Fig. 2 and Supplementary Fig. 2). Among all participants combined, plasma IgG Ab reactivity was found against a total of 372 *P. falciparum* Ag represented in a heatmap (Supplementary Fig. 2A and Supplementary Data 1). Overall, plasma anti-malarial IgG Ab from the older protected group reacted to a significantly higher number of *Pf* Ag (~177 antigens) compared to the susceptible group (~125) (Fig. 2A, upper graph). There were a greater number of IgG Ab reactive to *Pf*-Ag in the young protected compared to susceptible (160 vs 120), but this did not reach statistical significance. Further, plasma IgG Ab levels against individual *P. falciparum*-Ag were significantly higher in older protected plasma (~2) compared to that from susceptible (~0.7) and young protected participants (~0.95) (Fig. 2A, lower graph). There were higher *P. falciparum*-Ag IgG Ab levels in young protected compared to susceptible, but this was not statistically significant.

Of note, many of the *P. falciparum*-Ag reactive IgG Ab (*n* = 102) were detected in all groups but they exhibited significant differences in titers (Supplementary Fig. 2B and Supplementary Data 2). Among the 372 *Pf* reactive Ag a significantly higher IgG reactivity was measured against 45 *Pf* Ag in all protected compared to susceptible participant plasma which included 16 *Pf* surface proteins, 24 non-surface proteins and 8 unknown proteins (Fig. 2B and Supplementary Data 3). One of the most differentially abundant Ab reactivities was against the liver

stage antigen 3 (LSA-3), which is expressed during the liver stage and the asexual stage[56,57]. Unlike other Ags, LSA-3 is highly conserved and immunization with LSA-3 induces protection in primates against heterologous strain challenges[56]. Ab against the merozoite stage (MSP1, MSP4, MSP10, MSA180, and Duffy binding like merozoite surface protein[58]) were greater in older protected individuals compared to susceptible participants (Fig. 2B and Supplementary Data 3).

To determine if there were also functional differences in anti-merozoite Ab, we conducted a *P. falciparum* merozoite opsonization assay. *Pf* 3D7 strain was grown in vitro under standard conditions and merozoites were isolated and labeled with the YOYO-1 fluorescent dye to detect parasite nucleic acid by flow cytometry. Fluorescently labeled merozoites were incubated with plasma for 1 h, washed and co-cultured with THP-1 cells for 10 min prior to quantification of uptake by flow cytometry (Fig. 2C). Approximately 50% of THP-1 cells were YOYO-1⁺ when co-cultured with merozoites sensitized by preincubation with clinically immune participants plasma of all ages. In comparison, only 35% of THP-1 cells stained positive for YOYO-1 after co-culture with merozoites preincubated with plasma from the susceptible group. The addition of control plasma from malaria-unexposed US donors or no plasma, resulted in 10% YOYO-1⁺ staining of THP-1 cells. The *P. falciparum* merozoite uptake was reduced in the presence of anti-Fc receptor blocking Ab, indicating that plasma incubated *Pf* merozoites uptake involved Fc-receptor (Supplementary Fig. 2C). Uptake of merozoites preincubated with plasma from clinically immune participants was also partially blocked by cytochalasin D, indicating a role for actin polymerization. Overall, the older protected group's plasma had a greater number and abundance of antimalarial Ab exhibiting greater opsonization capacity of merozoites compared to that of the young susceptible group. The young protected Ab responses and merozoite opsonization were greater than the young susceptible; however, the differences were not statistically significant. The overall trend of increased Ab responses with age is consistent with prior reports showing breadth and intensity of Ab reactivity increases with age in participants living in malaria endemic regions[39].

### High-dimensional CyTOF analysis of immune cell populations

We next globally surveyed the immune cell populations occurring in the peripheral blood of participants undergoing acute *P. falciparum* malaria infection using a comprehensive immunophenotyping panel of markers we developed for mass cytometry (CyTOF) analysis of all major mononuclear myeloid and lymphoid cell lineages (Fig. 3, Supplementary Fig. 3A and Supplementary Data 4). The panel included established cell-surface and intracellular markers to discriminate effector and memory lymphocytes (CD45RA, CD27), and several helper CD4⁺ T cell functional subsets (FOXP3, CCR4, CCR6, PD-1, CXCR5, CXCR3). Using this panel, we analyzed PBMCs from clinically susceptible participants (*n* = 12) and the two clinically protected aged-matched (<13 yr, *n* = 9) and adult groups (≥13 yr, *n* = 10) (Fig. 1D and Supplementary Fig. 1E). We processed the analysis of our high dimensional mass cytometry data using both t-Distributed Stochastic Neighbor Embedding (t-SNE) and flow Self-Organizing Map (Flow-SOM) tools for unsupervised clustering, respectively allowing for dimensionality reduction, and for the visualization of discrete cell subsets and their relative proportions (Fig. 3A). The distribution of cell populations differed between the groups, with the highest cell subset diversity in the young protected group compared to the susceptible group, and as quantified by the Shannon diversity and equitability indexes, which measure how diverse and evenly distributed cell subsets in a population are. Further gating on known cell populations confirmed the heterogeneity in the proportions of immune cell subsets in the protected versus the susceptible groups (Fig. 3B and Supplementary Fig. 3B, C). Specifically, among innate

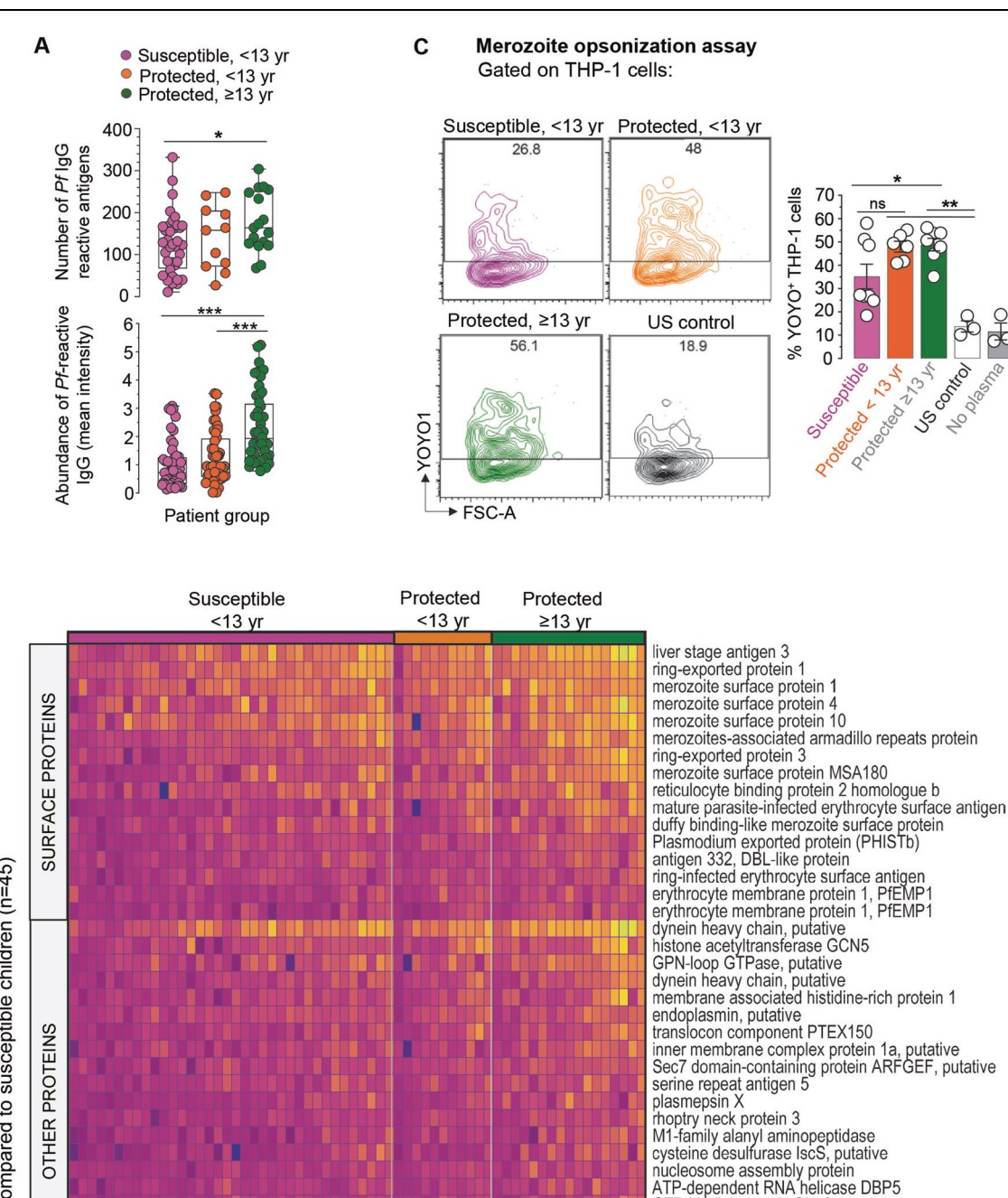

**A**

Legend:
- Susceptible, <13 yr
- Protected, <13 yr
- Protected, ≥13 yr

Number of *Pf* IgG reactive antigens (0–400), with * significance marker.

Abundance of *Pf*-reactive IgG (mean intensity) (0–6), with *** significance markers.

Patient group

**B**

IgG-specific *Pf* antigens with greater reactivity in protected compared to susceptible children (n=45)

Groups: Susceptible <13 yr, Protected <13 yr, Protected ≥13 yr

SURFACE PROTEINS:
- liver stage antigen 3
- ring-exported protein 1
- merozoite surface protein 1
- merozoite surface protein 4
- merozoite surface protein 10
- merozoites-associated armadillo repeats protein
- ring-exported protein 3
- merozoite surface protein MSA180
- reticulocyte binding protein 2 homologue b
- mature parasite-infected erythrocyte surface antigen
- duffy binding-like merozoite surface protein
- Plasmodium exported protein (PHISTb)
- antigen 332, DBL-like protein
- ring-infected erythrocyte surface antigen
- erythrocyte membrane protein 1, PfEMP1
- erythrocyte membrane protein 1, PfEMP1

OTHER PROTEINS:
- dynein heavy chain, putative
- histone acetyltransferase GCN5
- GPN-loop GTPase, putative
- dynein heavy chain, putative
- membrane associated histidine-rich protein 1
- endoplasmin, putative
- translocon component PTEX150
- inner membrane complex protein 1a, putative
- Sec7 domain-containing protein ARFGEF, putative
- serine repeat antigen 5
- plasmepsin X
- rhoptry neck protein 3
- M1-family alanyl aminopeptidase
- cysteine desulfurase IscS, putative
- nucleosome assembly protein
- ATP-dependent RNA helicase DBP5
- GTP-binding protein Obg1
- pentatricopeptide repeat-containing protein 1
- inner membrane complex protein 1i, putative
- alpha/beta hydrolase, putative
- protein TSSC1, putative

UNKNOWN:
- conserved Plasmodium protein
- probable protein
- conserved Plasmodium protein
- conserved Plasmodium protein
- conserved protein
- conserved Plasmodium protein
- conserved Plasmodium protein
- conserved Plasmodium protein

Normalized expression (-5, 0, +5)

**C** Merozoite opsonization assay
Gated on THP-1 cells:

Susceptible, <13 yr (26.8); Protected, <13 yr (48); Protected, ≥13 yr (56.1); US control (18.9)

YOYO1 vs FSC-A

% YOYO+ THP-1 cells (0–70): Susceptible, Protected < 13 yr, Protected ≥13 yr, US control, No plasma. Significance markers: ns, *, **

immune cell populations, we observed significantly higher frequencies of classical monocytes (CD14+CD16neg) in both the young and older protected groups (~22–27%) compared to the susceptible group (~7%, factor of ~3–4), consistent with prior observations[59]. However, non-classical, patrolling monocytes (CD14dimCD16+) were significantly reduced in the young protected group (~2%, factor of ~4) compared to the susceptible group (~8%). Frequencies of plasmacytoid dendritic cells (pDCs, CD123+, ~0.2%) and myeloid DCs (CD11c+, ~0.5%) were not significantly different across the groups. Likewise, the proportion of total B (~4-9%), CD4+ T (16-21%), CD8+ T (9-13%) and NK (~2%) lymphocytes did not significantly differ in all groups. Upon sub-analysis of these lymphocyte subsets into naïve (CD45RA+CD27+), effector (CD45RA+CD27−), and memory (CD45RA−CD27+/−) cells, there was an increased frequency of memory

**Fig. 2 | Monitoring of *P. falciparum* specific antibody responses in participants during mild malaria illness. A** Number of *Pf* antigens recognized by participant's plasma IgG (top) and abundance (bottom) of *P. falciparum*-reactive plasma IgG antibodies (mean intensity) in plasma determined on *Pf* antigen array. The center line of box plots denotes medians of data points and the box hinges correspond to the 1st and 3rd quartiles, the whiskers of the box and whisker plot extend from the box to the minimum and maximum observations within 1.5 times the IQR of the lower and upper quartile, respectively. Two-sided Welch's unpaired t-test was conducted between indicated groups (susceptible, <13 yr (*n* = 36, pink), protected, <13 yr (*n* = 11, orange) and protected, ≥13 yr (*n* = 17, green) *p < 0.05, **p < 0.01, ***p < 0.001. **B** Heatmap of normalized levels of IgG antibodies in individual participants (columns) against 45 *Pf* antigens (rows) that are significantly higher in protected

versus susceptible groups (Two-sided Mann–Whitney test, p < 0.0001. **C** *Pf* merozoite opsonization assay with YOYO-1 labeled *Pf* merozoites incubated with indicated participant's plasma, washed and then co-cultured with THP-1 cells. FACS plots of a representative sample within indicated groups, quantifying YOYO1+ THP1 cells after the co-culture (left). Summary of %YOYO1+ THP1 cells (right) indicating % *Pf* merozoite uptake when labeled merozoites were incubated with or without indicated plasma (*n* = 3) from susceptible (*n* = 8, pink), protected <13yr (*n* = 6, orange), protected ≥13 yr (*n* = 7, green) and co-cultured with THP1 cells. Two-sided Welch's unpaired *t* test were conducted between indicated groups, *p < 0.05, **p < 0.01, ***p < 0.001. Pool of two independent experiments represented as individual data and means ± SEM (**C**).

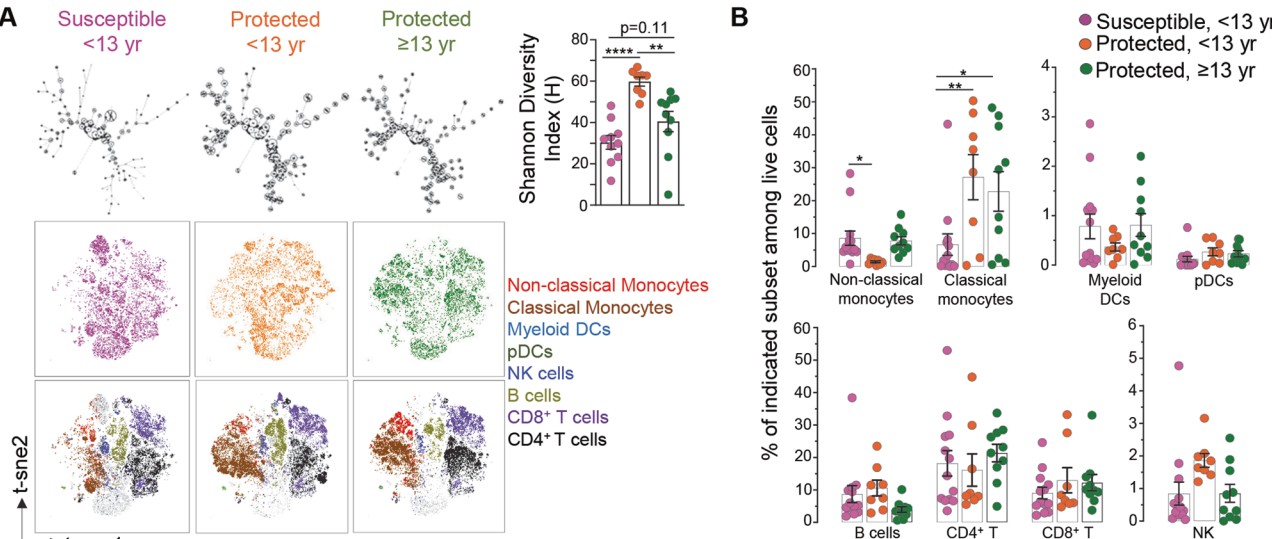

**Fig. 3 | Overview of CyTOF analysis of immune cell populations in the blood of malaria-infected participants. A** FlowSOM (top) and t-SNE (bottom) visualization of live PBMCs of a representative sample within each indicated group based on lineage markers expressed on cells and detected by mass cytometry (CyTOF). Shannon Diversity Index for each group of patients (susceptible, <13 yr (*n* = 10, pink), protected, <13 yr (*n* = 8, orange) and protected, ≥ 13 yr (*n* = 10, green), with individual symbol representing one patient. Overlaid immune cell populations on the t-SNE plots are indicated. **B** Summary of innate and adaptive immune cell frequencies in PBMCs of malaria infected participants. Mean cell frequency is

indicated for susceptible, <13 yr (*n* = 13, pink), protected, <13 yr (*n* = 8, orange) and protected, ≥ 13 yr (*n* = 10, green). Summary of the immune cell subset frequency in participants during malaria illness among indicated groups are shown. Bar graphs represent mean cell frequency values. Statistical significance is calculated by two-sided unpaired Student's *t* test between indicated groups with correcting for multiple comparisons, *p < 0.05, **p < 0.01, ***p < 0.001, ****p < 0.0001. Pool of three independent experiments represented as individual data and means ± SEM (**A**, **B**).

T cells among CD4+ T but not CD8+ T cells, in the older protected group compared to the susceptible group, which is likely age-related (-33 versus 20%, factor of -1.6 fold, Fig. 4A, B)[60].

Interestingly, the proportion of circulating follicular helper CD4+ T (cT$_{FH}$) cells, but not of FOXP3+ regulatory T (T$_{reg}$) cells (~4-6%) among CD4+ T cells, was increased in both protected groups compared to the susceptible counterpart (-8 versus 12–17%). Consistent with this observation, we detected a significantly higher frequency of memory B cells (CD19+CD27+, factor of -2-3) among both protected compared to the susceptible group (21–31% versus 12%, Fig. 4C). We also quantified increased proportions of immunoglobulin isotype class-switched B cells (39 versus 21–24% of IgM−IgD− B cells, factor of -1.8) and reduced frequency of non-switched B cells when comparing adult protected to the two other groups (42 versus 55-58% of IgM+IgD+ B cells, factor of -1.3). Of note, similar frequencies of atypical memory B cells (CD27−CD21−), which were reported to be expanded in malaria-infected patients, were measured across all groups[10,61,62]. In summary, both clinically protected groups undergoing an episode of mild *P. falciparum* malaria infection exhibited a significantly expanded population of classical monocytes, CD4+ cT$_{FH}$ and class-switched B cells compared to the clinically susceptible group.

## Single-cell transcriptomic analysis of circulating memory CD4+ T cells in protected participants

Since CD4+ T cells are essential to the control of *P. falciparum* infections and the induction of long-term protective parasite-specific humoral responses[43], we next sought to further characterize the memory CD4+ T cells present in the clinically protected adult group during an episode of mild malaria, using single-cell transcriptomic and T cell receptor (TCR) sequencing. Utilizing high-speed Fluorescence Activated Cell Sorting with gating on live cells, memory CD4+ T cells (CD45RO+ CD27+ or CD27−) and naïve CD4+ T cells (CD45RO−CD27+) were purified from the PBMCs of three participants in the clinically protected adult group (age 18, 25, 34 yrs, Supplementary Fig. 4A, B). Whole transcriptome single-cell RNA sequencing was performed on the sorted populations using the 10X Genomics platform. We integrated the single cell transcript expression data from all three participants and conducted dimensional reduction analysis to visualize transcriptionally distinct cell subsets in a UMAP plot based on transcriptional similarities between cells (Fig. 5A). This identified eight transcriptionally distinct memory CD4+ T cell subsets, found across the three participants analyzed. The distribution of subsets within individual participants was similar (Supplementary Fig. 4C). As a control, we

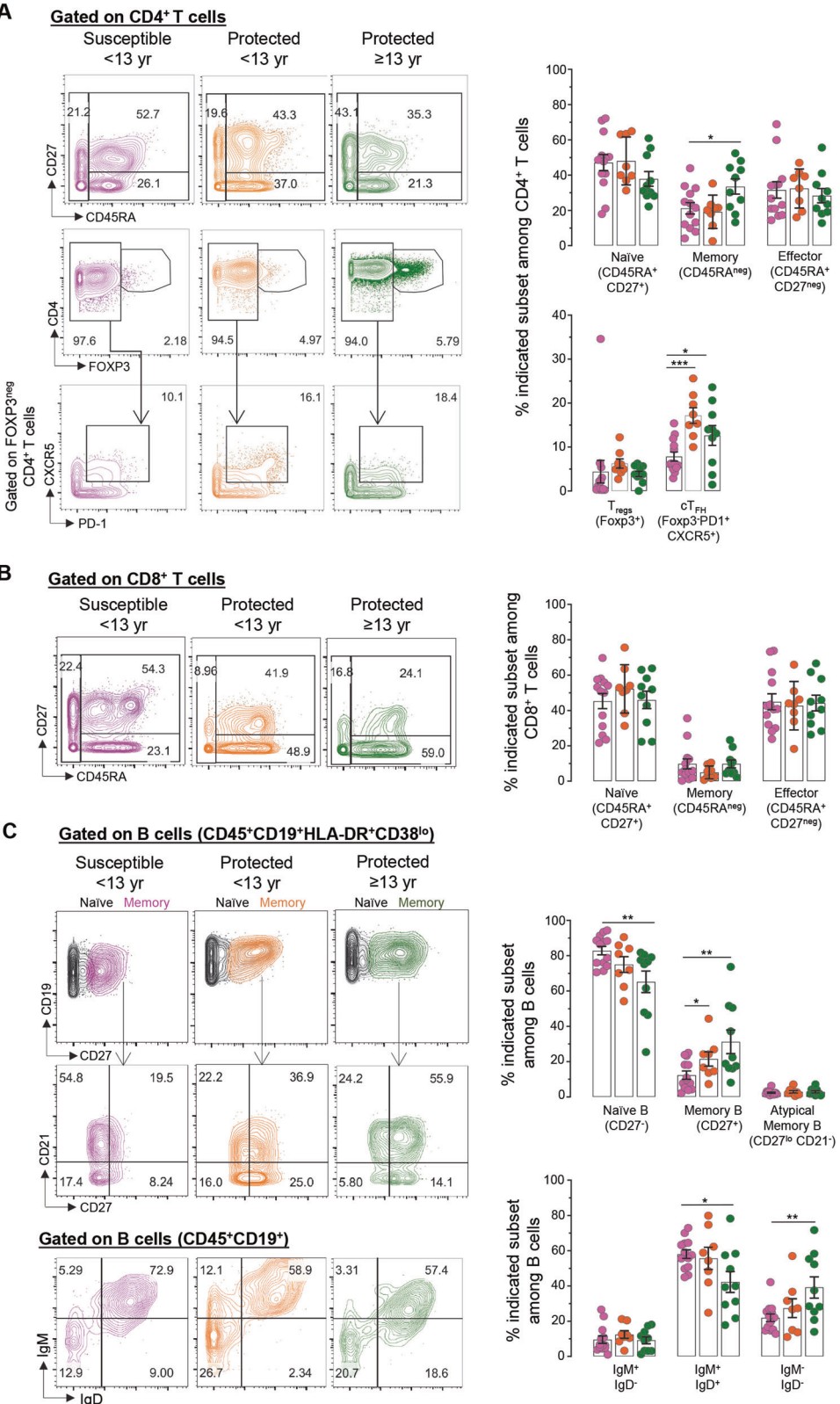

**Fig. 4 | CyTOF analysis of T and B cell populations in the blood of malaria infected participants. A–C** Representative CyTOF plots of one participant from each group stained with a 33-marker heavy metal tagged antibody cocktail. Contour plots indicate the staining and gating strategy for CD4+, CD8+ T cell and B cell subsets among live PBMCs. Summary of the immune cell subset frequency in participants during malaria illness among susceptible, <13 yr ($n = 13$, pink), protected, <13 yr ($n = 8$, orange) and protected, ≥13 yr ($n = 10$, green) are shown. Bar graphs represent mean cell frequency values. Statistical significance is calculated by two-sided unpaired Student's $t$ test between indicated groups, *$p < 0.05$, **$p < 0.01$, ***$p < 0.001$. Pool of three independent experiments represented as individual data and means ± SEM (**A–C**).

**A**  **Single cell RNA-seq analysis of memory CD4⁺ T cell reveals 8 clusters (n=3 protected participants)**

**B**  **Expression levels of indicated genes by memory CD4⁺ T cell cluster**

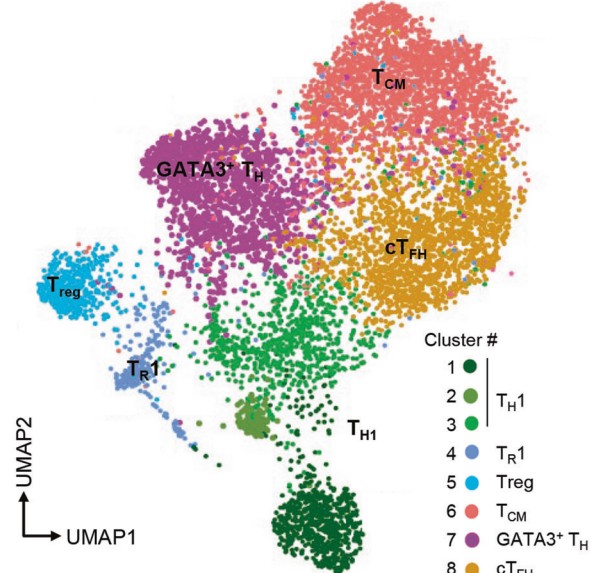

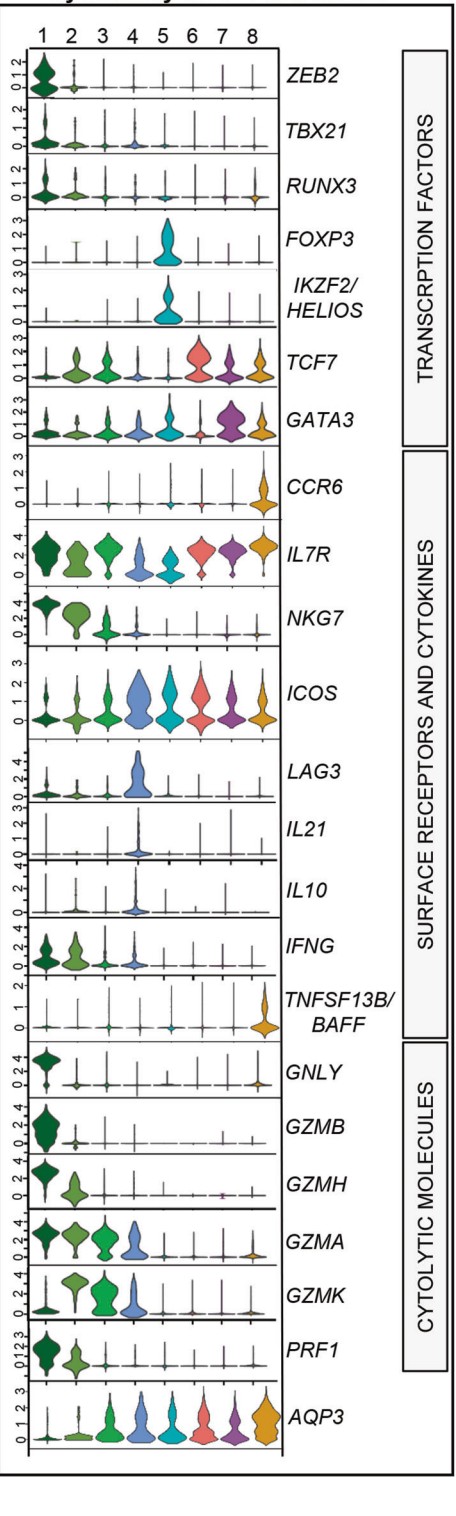

**C**  **Genes expressed in each memory CD4⁺ T cell cluster**

| Cluster | Predicted T$_{subset}$ | Signature genes |
|---|---|---|
| 1 | | NKG7, GNLY, GZMH, CCL5, CCL4, GZMB, GZMA, PRF1, IFNG, GZMM, KLRG1, ZEB2, CX3CR1, TNF, IL2RG, LY6E, TBX21, GBP5, IL32, RUNX3, CCL3 |
| 2 | T$_H$1 | GZMK, CCL5, CCL4, GZMA, NKG7, IFNG, GZMM, TNF, CD27, KLRG1, EOMES, IL2RB, PDCD1, PECAM1, CXCR4 |
| 3 | | GZMK, GZMA, CCL5, CXCR3, GZMM, IFNG, IL32, HOPX, KLRG1, SAMD3, GBP5 |
| 4 | T$_R$1 | LAG3, CTLA4, CD59, IL10, CD38, IL32, GZMA, IL2RB, GZMK, CD27, IL21, TIGIT, PRDM11, BATF1, CXCR6, GZMM, MKI67, ICOS, LY6E, IRF4, ID2 |
| 5 | T$_{reg}$ | FOXP3, CTLA4, IKZF2, SELL, IL32, CD27, TIGIT, BATF, IL2RA, PRDM1, IL10RA, CCR4, GATA3, ICOS |
| 6 | T$_{CM}$ | TCF7, LEF1, CCR7, SELL, IL6ST, AREG, CD27, ICOS |
| 7 | GATA-3⁺ T$_H$ | GATA3, LGALS1, TNFSF10, CCR10, SOCS3, LMO4, IL13 |
| 8 | cT$_{FH}$ | IL7R, CCR6, LTB, AQP3, TNFSF13B, CMTM6, LGALS3, TOB1, ZFP36 |

**Fig. 5 | Single-cell transcriptomic analysis of memory CD4⁺ T cells from protected participants during malaria illness. A** UMAP visualization of integrated single-cell RNA-seq results on memory CD4⁺ T cells isolated from 3 protected participants (age 18.4, 24.6, 34 yrs) during malaria infection. Clusters of memory CD4⁺ T cells with distinct transcriptional profiles are colored as indicated in the legend. **B** Normalized expression levels of indicated genes within each distinct memory CD4⁺ T cell cluster defined in (**A**). **C** Table of top signature genes expressed in each distinct memory CD4⁺ T cell cluster defined in (A).

examined naïve CD4$^+$ T cells which exhibited a very different transcriptional profile and, as expected, clustered with central memory CD4$^+$ T cells visualized on the UMAP plots (Supplementary Fig. 4D and Supplementary Data 5). None of the other, more differentiated clusters of memory CD4$^+$ T cells overlapped with that of the naïve.

Among clusters of memory CD4$^+$ T cells, clusters 1, 2, and 3, showed a robust T$_H$1 expression profile (Fig. 5B, C). Cluster 1 compared to 2 and 3, exhibited significantly higher levels of transcripts coding for T-BET (*TBX21*), the master Th1 transcriptional regulator, and for ZEB2, a key transcription factor (TF) that cooperates with T-BET to drive terminal effector CD8$^+$ T cell differentiation[63,64]. Moreover, cluster 1 expressed higher levels of transcripts coding for the TF RUNX3, which epigenetically programs and enables the differentiation of long-lived cytolytic memory T cells[65,66]. We also noted upregulation in cluster 1 of genes encoding multiple effector molecules including granzymes (*GZMB, GZMA, GZMK, GZMH*), granulysin (*GNLY*), perforin (*PRF1*), IFNγ, as well as *NKG7* which codes for a membrane protein that regulates target cell killing[67]. Clusters 2 and 3 selectively expressed transcripts coding for granzyme K (*GZMK*) but not B (*GZMB*), compared to cluster 1. Other notable differences between cluster 2 and 3 were the abundance of granzyme H, IFNγ and perforin-encoding transcripts in cluster 2 (Fig. 5B). While these 3 clusters have a clear T$_H$1 cell signature, they may represent distinct stages or paths of differentiation.

In addition, the single cell transcriptomic analysis identified a subset of effector and regulatory T cells, likely T$_R$1 cells (cluster 4), with migratory and proliferating capacities (CXCR6, KI67) that expressed transcripts coding for inhibitory receptors (LAG-3, CTLA-4, ICOS, TIGIT) and the cytokines IL-10 and IL-21, while also encoding mRNA of TF that regulate terminal effector cell differentiation (*ID2, IRF4*)[46,47,68,69]. Another cluster (cluster 5) included T$_{reg}$ cells that expressed increased transcript levels of the TFs FOXP3 and IKZF2 (Helios), and inhibitory receptors CTLA-4, ICOS, and TIGIT. Among the largest subsets revealed by this analysis, we found gene expression signatures associated with central memory cells (T$_{CM}$, cluster 6) and GATA-3$^+$ (possibly T$_H$2 effector cells, cluster 7) with high levels of transcripts coding for their associated transcription factors TCF7 and GATA-3, respectively. Lastly, we report an abundant subset of memory CD4$^+$ T cells (cluster 8) that contained high levels of transcripts coding for the TF TCF7, the chemokine receptor CCR6, the cytokine BAFF, and the IL-7 receptor, consistent with a signature of circulating T$_{FH}$ cells (cT$_{FH}$)[9,68,69].

## Most expanded memory CD4$^+$ T cell clones expressing shared TCR clonotypes have a cytolytic transcriptional signature

Concomitantly with the single cell transcriptomic analysis, we sequenced the TCRα and TCRβ variable chains expressed in each memory CD4$^+$ T cell sorted from *P. falciparum*-infected protected participant PBMC (Fig. 6 and Supplementary Fig. 5). With this approach, we could determine if individual memory CD4$^+$ T cells expressed exactly the same pair of TCRα and β chain sequences and thus shared the same TCR clonotypes (Fig. 6A and Supplementary Fig. 5A). Between ~0.3% to 4.3% of memory CD4$^+$ T cells bear identical TCR clonotypes in each participant, suggesting that each of these expanded T cell clones are the progeny of the same original clone proliferating in response to a specific Ag. These frequencies of pathogen-specific CD4$^+$ T cells are consistent with those reported in patients undergoing influenza infection[70]. Since the TCR sequencing analysis was conducted on the memory CD4$^+$ T cells for which we ran our transcriptomic analysis, we could link each expanded T cell clone to its expression signature and the memory CD4$^+$ T cell clusters defined in Fig. 5A (Fig. 6B). The large majority (~95%) of expanded T cell clones sharing a same TCR clonotype across the three participants belonged to cluster 1 -which expressed transcripts encoding a robust cytolytic T$_H$1 effector program (Fig. 6C). Most expanded clones expressed high levels of transcripts coding for master transcription factors governing effector T cell signatures and fates, namely ZEB2

(~60%), RUNX3 (~35%) and T-BET (~20%) (Supplementary Fig. 5B). These cells also expressed transcripts coding for the cytolytic effector proteins NKG7 (~100%), granulysin (*GLNY*, ~93%), granzymes (*GZMH, GZMA*, ~94%), granzyme B (*GZMB*, ~70%) and perforin (*PRF1*, ~80%). Clusters 2 and 3 (T$_H$1, GMZK$^+$, GZMB$^-$GNLY$^-$), and cluster 4 (T$_R$1) respectively accounted for ~2% of the expanded clones (Fig. 6C).

To further determine if expanded T cells expressing the same TCR clonotypes were found in the multiple memory CD4$^+$ T cell clusters defined in Fig. 5A, we next tracked the fate of each of them across the various clusters (Fig. 6D). Several clones sharing the same TCR clonotype, belonged to various clusters (1 and 2, 1 and 3, 1 and 4, 1, 2 and 3, 2 and or 4, 1, 3 and 4), suggesting that an expanded clone can have multiple T$_H$ fates. To assess individual clone fates in the context of T-cell differentiation, we defined single-cell trajectories using Monocles[71]. Monocles models single-cell gene expression data as a function of pseudotime to place a cell along the differentiation process (Fig. 6E). Pseudotime is an arbitrary measure reflecting how far an individual cell is in the differentiation process with respect to a predefined cell of origin (roots). Here we used T$_{CM}$ as the root (cluster 6, Fig. 5A) since these cells are reported to maintain the highest differentiation potential among memory cells. Together with their individual distribution among the relevant clusters (1 to 4, Fig. 6E), this analysis suggested that the most highly differentiated memory CD4$^+$ T cell subset (T$_H$1, cluster 1), where expanded clones accumulated, likely derived from less differentiated states that included T$_R$1, T$_H$1 cluster 2 or T$_H$1 cluster 3. Taken together these results show that the great majority of expanded memory CD4$^+$ T cell clones sharing identical TCR clonotypes in clinically protected individuals during mild *P. falciparum* malaria infection, were terminally differentiated and expressed a robust cytolytic effector program.

To identify potentially unique biomarkers of the expanded clones, we defined the clonotype transcriptional signatures and compared their proportion among all memory CD4$^+$ T cell transcripts (Fig. 6F and Supplementary Fig. 5C). Among the transcripts encoding for transcription factors, *ZEB2* was the most highly expressed in expanded clones, and ~35% of the *ZEB2*-expressing memory CD4$^+$ T cells were expanded clones. In addition, among the transcripts coding for cytolytic effector molecules, *GZMB* expressing memory CD4$^+$ T cells consisted of the largest proportion of expanded clones among memory CD4$^+$ T cells (~60%). Both *ZEB2*- and *GZMB*-expressing memory CD4$^+$ T cells contained a significantly higher proportion of expanded clones compared to all memory CD4$^+$ T cells (factor of 7–12, Fig. 6G). These results collectively suggested that a substantial proportion of memory CD4$^+$ T cells that express *ZEB2* and/or *GZMB* transcripts are clonally expanded and presumably respond to *P. falciparum* Ag, raising the possibility that expression of ZEB2 and perhaps granzyme B, may be useful surrogate markers to identify malaria-specific memory CD4$^+$ T cells that are clonally expanded.

## ZEB2$^+$ memory CD4$^+$ T cells recognize the *P. falciparum* derived circumsporozoite protein (CSP) antigen

Given the importance of TF in orchestrating T cell fates, we focused on the ZEB2 TF and first confirmed that we could detect a population of ZEB2$^+$ memory CD4$^+$ T cells in PBMC from malaria-infected patients by flow cytometry (Fig. 7A). We used PBMC from clinically immune adult participants from our cohort that were undergoing a malaria episode (Supplementary Fig. 6A). Consistent with the proportion of expanded clones among memory CD4$^+$ T cells in our single cell analysis (between 0.3 and 4.3%, Fig. 6A), ZEB2$^+$ memory CD4$^+$ T cells (~half of expanded clones, Fig. 6G) represented between 0.5 and 10% of all memory CD4$^+$ T cells. To next assess if ZEB2$^+$ memory CD4$^+$ T cells reacted to *P. falciparum* Ag, we conducted an in vitro Activation Induced Marker (AIM) assay which monitors the upregulation of several cell surface activation markers (here CD25, OX40 and 4-1BB) in response to given Ag[72], allowing for the identification of cognate Ag-reactive T cells (Fig. 7B).

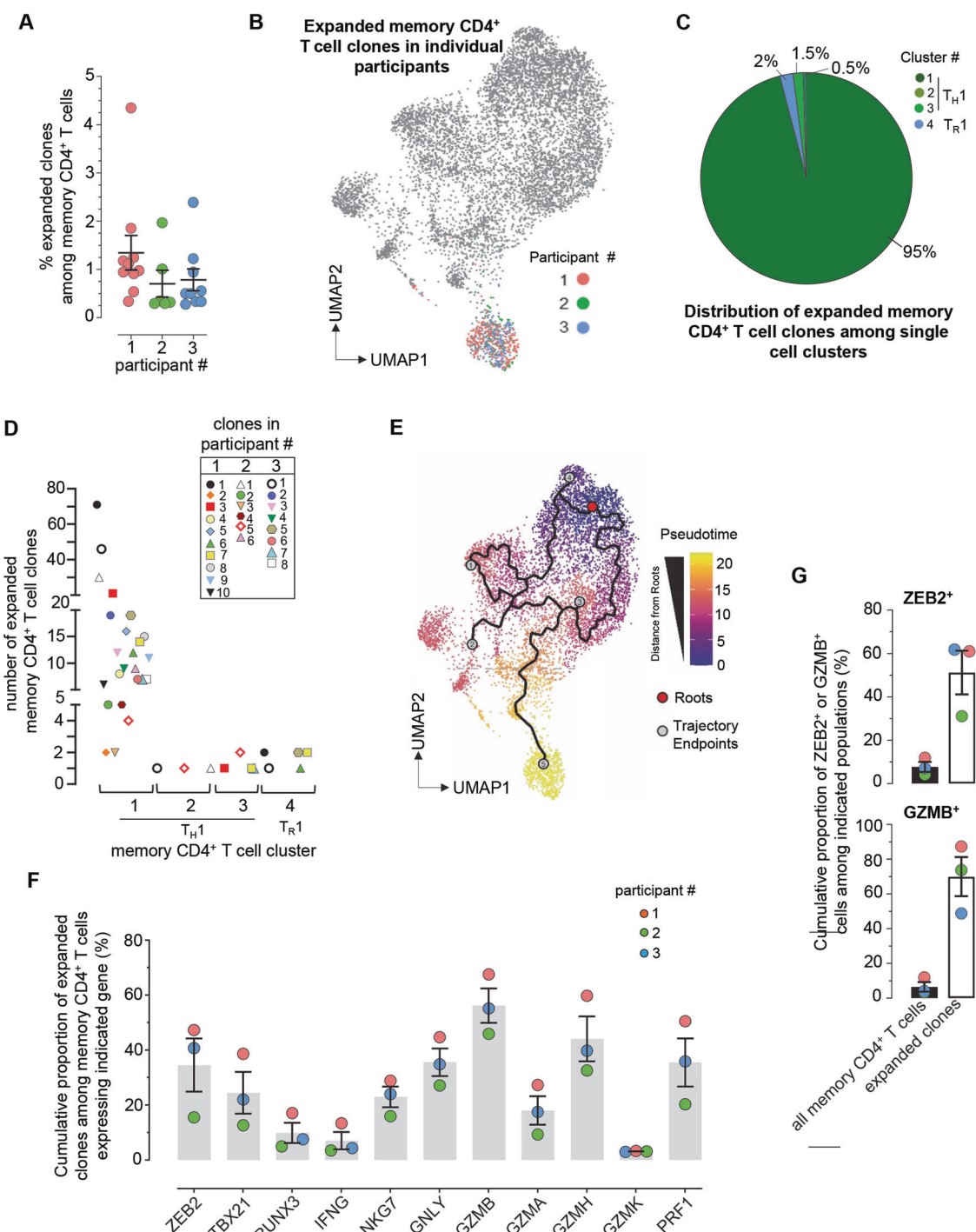

**Fig. 6 | Expanded memory CD4⁺ T cell clones in malaria protected patients exhibit a cytolytic gene expression signature. (A)** Frequency of each expanded memory CD4⁺ T cell clone expressing an identical TCR clonotype among all memory CD4⁺ T cells in patient 1 (*n*=10, Orange), patient 2 (*n*=6, Green), patient 3 (*n*=9, Blue), as defined by single-cell TCR-seq. **(B)** Expanded memory CD4⁺ T cell clones (at least 10 cells sequenced with the exact same TCRα and TCRβ chains) in each of the 3 participants, highlighted on the concatenated UMAP visualization shown in Fig. 5A. **(C)** Pie-chart of expanded memory CD4⁺ T cell clone proportions across distinct transcriptional clusters defined in Fig. 5A and across the 3 protected

participants. **(D)** Distribution of individual clones with identical TCR clonotype across memory CD4⁺ T cell cluster 1, 2, 3 and 4 (as defined in Fig. 5) and across the 3 protected participants. **(E)** Pseudotime analysis of the memory CD4⁺ T cell subsets of Fig. 5A. **(F)** Cumulative proportion of expanded memory CD4⁺ T cell clones expressing indicated gene-encoding transcripts across the 3 participants.
**(G)** Proportion of cells that express *ZEB2* or (Granzyme B (*GZMB)* transcripts among all memory CD4⁺ T cells or only among expanded T cell clones (n=3). Pool of three independent experiments represented as individual data and means ± SEM (A, F, G).

We reasoned that PBMCs from participants in our study cohort contained circulating monocytes, dendritic cells and *P. falciparum*-specific B cells that could recognize and present *P. falciparum* Ag to the T cells. We used major known *P. falciparum* Ag, namely the circumsporozoite-

derived protein (CSP), the merozoite-derived protein 1 (MSP-1), the apical membrane antigen-1 (AMA-1) and the erythrocyte binding antigen 175 (EBA-175), all of which against we detected IgG Ab reactivity in patient sera (Supplementary Fig. 2A and Supplementary Data 1). ZEB2⁺

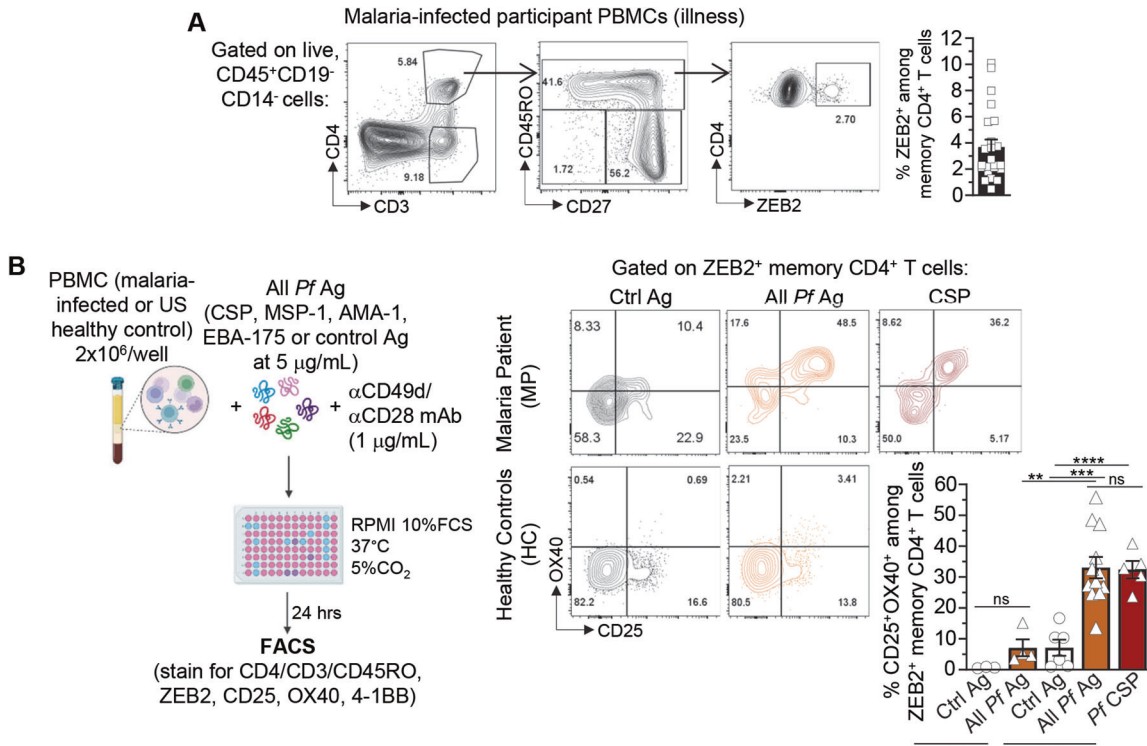

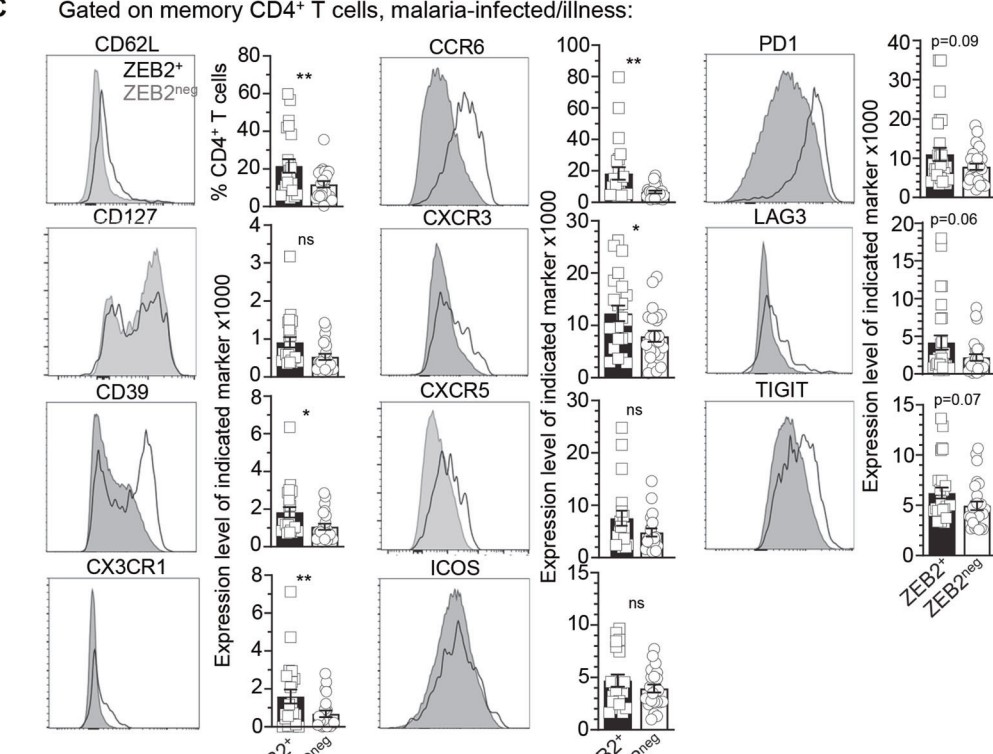

**Fig. 7 | ZEB2⁺ memory CD4⁺ T cells react to the *P. falciparum*-derived circumsporozoite protein 1 (CSP) antigen. A** Gating strategy of ZEB2⁺ memory CD4⁺ T cells on a representative protected malaria participant during illness. Frequency of ZEB2⁺ memory CD4⁺ T cells across all study participants (n=23) and during a malaria episode. **B** Experimental design of the AIM assay. Representative FACS dot plots of CD25 and OX40 expression by ZEB2⁺ memory CD4⁺ T cells in i) Healthy controls (HC) stimulated with control antigen (ctrl Ag) (*n* = 3,white) or all *Pf* Ag (Circumsporozoite Protein (CSP), Merozoite Surface Protein-1 (MSP-1), Apical

Membrane Antigen 1 (AMA-1), Erythrocyte Binding Antigen 175 (EBA-175) (*n* = 4,brown) and in ii) Malaria Patients (MP) stimulated with ctrl Ag (*n* = 6,white), all *Pf* Ag (*n* = 12, brown) or *Pf* CSP (*n* = 5, red). **C** Cell-surface expression of indicated markers by ZEB⁺ versus ZEB2ⁿᵉᵍ memory CD4⁺ T cells across participants (*n* = 23) using spectral flow cytometry. Each symbol is one participant. *P*-values were calculated using two-sided unpaired Student's *t* tests. Pool of 3 independent experiments represented as individual data and means ± SEM (**A**–**C**).

memory CD4+ T cells from malaria-patients responded to a mixture of the four *P. falciparum* Ag and CSP alone (~35% CD25+OX40+, ~16% CD25+4-1BB+) while control Ag (i.e., the Exo protein A (EPA) and the *P. pastoris* (*Pp*) Pfs25M) stimulation failed to induce any significant upregulation of these activation marker (Fig. 7B and Supplementary Fig. 6B). In addition, *P. falciparum* or control Ag-stimulated ZEB2+ memory CD4+ T cells from healthy US participants also failed to exhibit any significant Ag-specific activation. Of note, ZEB2neg memory CD4+ T cells from malaria-infected patients also contained *P. falciparum*-Ag reactive T cells, although in lower proportion among the memory CD4+ T cells (~15%, Supplementary Fig. 6C), confirming that ZEB2+ memory cells were enriched in *P. falciparum*-specific memory CD4+ T cells compared to non-ZEB2+ counterparts.

Further phenotyping of ZEB2+ versus ZEB2neg memory CD4+ T cells using high dimensional spectral flow cytometry and markers chosen from our single cell analysis (Supplementary Fig. 6D and Supplementary Data 4), revealed that while ZEB2+ memory CD4+ T cells were largely effector memory T (T$_{EM}$) cells (CD62LloCX3CR1loCD127lo), they contained a significantly higher proportion of CD62L+CX3CR1+CD127+ cells compared to ZEB2neg counterparts. A significant higher proportion of them also expressed CD39 which is associated with Ag-experienced T cells. Compared to the ZEB2neg cells, we also noted higher expression of the co-inhibitory receptors PD-1, LAG-3, and TIGIT as well as of the CCR6 and CXCR3 chemokine receptors involved in cell recruitment to inflammatory tissue sites. While trending higher, cell surface expression of CXCR5 and ICOS were not significantly different from that of ZEB2neg cells. In summary, ZEB2+ memory CD4+ T cells recognized the *P. falciparum*-derived CSP Ag, exhibited a T$_{EM}$ cell phenotype, and expressed markers associated with exhaustion and higher levels of chemokine receptors mediating cell homing to inflammatory tissues.

### Expansion of ZEB2+ memory CD4+ T cells in patients gaining clinical immunity

Since ZEB2+ memory CD4+ T cells contained clonally expanded *P. falciparum* CSP-specific T cells, we next hypothesized that protected patients should have greater proportion of these cells than susceptible ones, if these cells are indeed associated with clinical immunity against malaria. Thus, we tracked ZEB2+ cells among memory CD4+ T cells in the blood of our study participants at the time of acute infection and 30 days later at convalescence (Fig. 8A). Both in susceptible and young protected groups, ZEB2+ memory CD4+ T cells were significantly expanded during the infection compared to convalescence. However, we did not find any significant differences in their proportions across these two groups and times. ZEB2+ cells were also detected in older and clinically protected participants at equivalent frequencies as in other convalescent participants, but we did not detect a greater expansion of the ZEB2+ memory CD4+ T cells during illness in this older protected group, which contrasted with our observation in the younger groups.

In order to further refine this analysis, we next asked if we could identify participants among the susceptible group who were gaining (or predicted to gain) clinical immunity or not during the course of our 18-month longitudinal study, and if the expansion of ZEB2+ memory CD4+ T cells differed between these two groups. To find such participants, we first identified those who had at least two episodes of clinical malaria. We then determined each individual rate of recurrent clinical malaria and using a mixed effects model, we identified five participants who were gaining clinical immunity and four participants who remained clinically susceptible during the study and had available PBMC (Supplementary Fig. 6E). We compared the frequency of ZEB2+ memory CD4+ T cells during the enrollment episode of clinical malaria to the frequency during a subsequent episode of clinical malaria between the two groups. The ZEB2+ memory CD4+ T cells in these two groups expanded only in the blood of those gaining clinical immunity

but not in the other group, consistent with our original hypothesis that expansion of these cells may correlate with the acquisition of protective immunity against clinical malaria (Fig. 8B). Further analysis of the phenotype of ZEB2+ memory CD4+ T cells revealed an increased expression of inhibitory receptors (PD-1, LAG-3 and TIGIT) and chemotactic receptors (CXCR3, CCR6) on the CD39+ Ag-activated sub-population from susceptible compared to protected aged-matched patients (Fig. 8C). In addition, ZEB2+ memory CD4+ T cells from protected individuals had significantly lower proportions of PD-1+LAG-3+ cells compared to the susceptible group (Fig. 6D). Thus overall, these data suggested that ZEB2+ memory CD4+ T cells may expand more in individuals who develop clinical immunity against *Pf* malaria compared to those that do not. Results also reveal that activated (CD39+) ZEB2+ memory CD4+ T cells from protected participants express lower levels of inhibitory and chemotactic receptors than those from susceptible patients.

## Discussion

In this work, we conducted a longitudinal study to monitor individuals living in a malaria-endemic area with high, year-round transmission, where residents develop naturally acquired clinical immunity against severe disease and clinical illness during *P. falciparum* infection. We postulated that a comparison between individuals who have or have not yet developed clinical immunity would reveal new functional immune cell subsets associated with protection that may ultimately inform the design of an efficacious malaria vaccine. We present a detailed phenotypical classification of study participants who vary in clinical immunity to provide a robust foundation for our immune analyses. During an episode of mild malaria, we found that protected individuals were less likely to be febrile, had lower blood parasite loads, and higher hematocrit. These individuals exhibited greater parasite-specific Ab responses (breadth, titers, opsonization) and higher proportion of classical monocytes, switched IgG+ B cells and circulating CD4+ T$_{FH}$ cells. Most importantly, we report the identification of a subset of highly differentiated memory CD4+ T cells that are clonally expanded during clinical malaria, recognize the *P. falciparum*-derived CSP pre-erythrocytic stage Ag, express the ZEB2 transcriptional regulator and exhibit a robust cytolytic effector gene signature. These cells exhibit greater expansion in infected individuals that may be gaining clinical immunity over the course of the study than those that did not. Furthermore, age-matched protected individuals express lower levels of T cell exhaustion markers (PD-1, LAG-3, TIGIT) and chemotactic receptors involved in homing to inflammatory sites (CXCR3, CCR6).

The presence of cytolytic CD4+ T cells in malaria-infected patients and mouse models of malaria was first reported several decades ago[40,73]. Building on the hypothesis that CD8+ T-cell dependent cytolytic activity could play a major role in anti-sporozoite immunity, these original studies isolated MHC class II-restricted cytolytic CD4+ T cell clones that recognized a *P. falciparum* CSP derived epitope (human) and a *P. berghei* epitope common to both the liver and blood stage infection of this rodent parasite. In both studies, these clones could expand, produce IFNγ and effectively lyse parasite Ag-pulsed target cells. The mouse model further established that transfer of these cells protected mice against infection with a live sporozoite challenge. In these reports, the CD4+ T cell cytolytic clones were isolated from the animal model of malaria or human participants that received irradiated sporozoite immunizations. The presence of parasite (CSP)-specific CD4+ T cells with cytolytic activity was more recently confirmed in individuals inoculated with irradiated or live sporozoite immunizations[41,42,74]. To our knowledge, however, no studies have reported the presence of such cells in endemic area residents who have developed immunity against *P. falciparum*. This finding is particularly interesting as sporozoite immunization induces potent protection against homologous *P. falciparum* challenge infections and is viewed

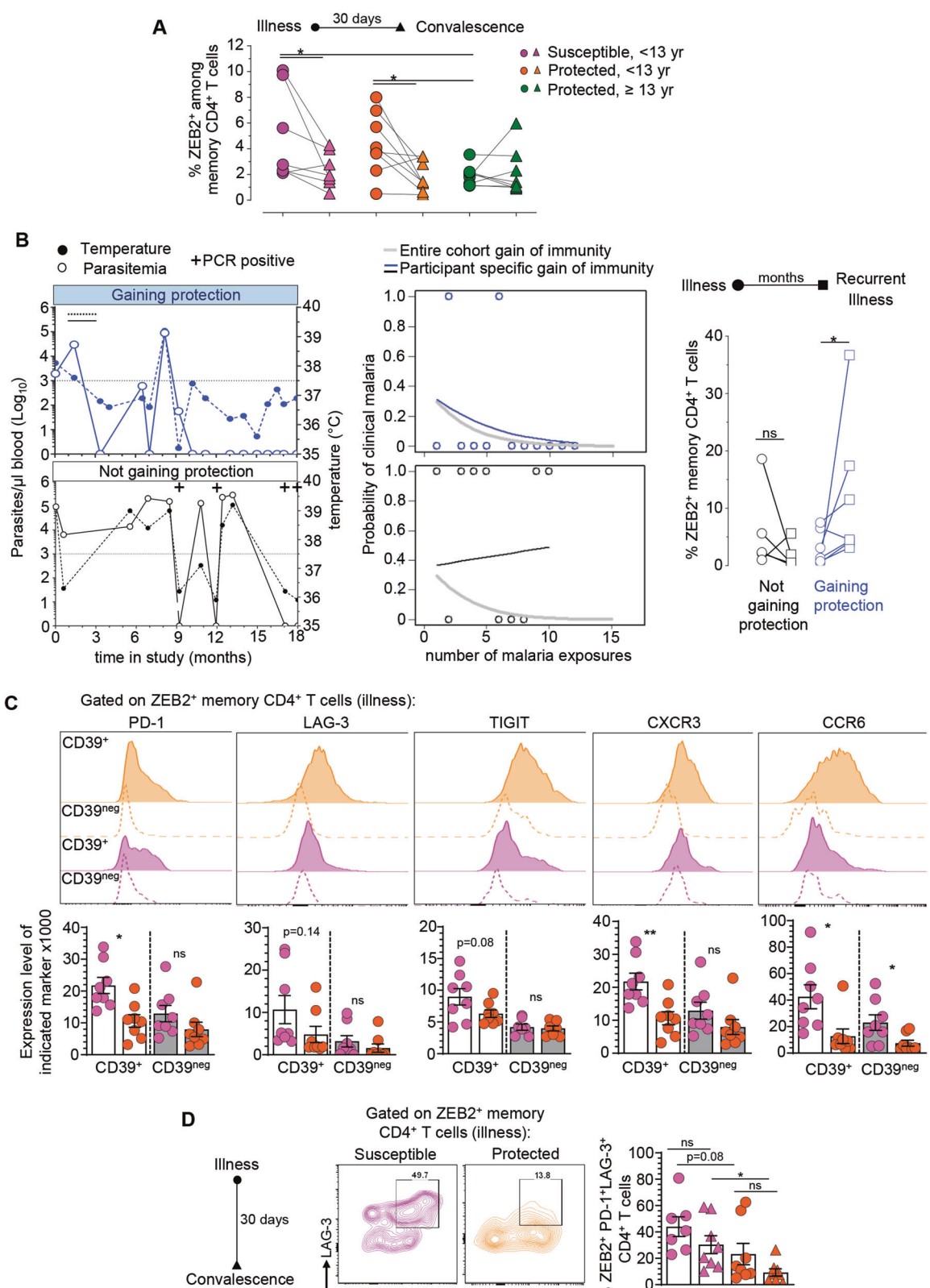

as a promising vaccine strategy that could induce high levels of immunity[25,27,28,75,76]. While immunity is not sterilizing, it still effectively lessens the symptoms of clinical malaria episode and prevents severe disease, and it is long-lived. The ZEB2+ cytolytic memory CD4+ T cells we described are clonally expanded in the blood of clinically immune individuals and react to the *P. falciparum*-derived CSP Ag. We also confirmed their overall expansion among memory CD4+ T cells in the

blood of susceptible and young protected groups undergoing an episode of acute malaria compared to convalescence. This was not observed in the older clinically immune group which may be the result of differences in the kinetics and localization of these cells. The immune response in vaccinated or immune individuals is known to be faster and more potent upon infection, likely enabling more rapid containment of microbial pathogens which could involve the ZEB2+

**Fig. 8 | ZEB2+ memory CD4+ T cells expand during *P. falciparum* infection and in individuals gaining clinical immunity. A** Frequencies of ZEB2+ cells among memory CD4+ T cells in susceptible, <13 yr (n = 7, pink), protected, <13 yr (n = 8, orange) or protected, ≥13 yr (n = 8, green) malaria participants during illness (circle symbol) and at day 30 post illness (convalescence, triangle symbol).
**B** Representative examples of recorded temperature, blood parasitemia and *Pf* PCR testing of two aged-matched participants either gaining clinical immunity or not during our 18-month study. Predictive model of such behavior in each participant is shown. Frequencies of ZEB2+ memory CD4+ T cells across participants gaining (n =

6) or not (n = 6) clinical immunity during the study period, at an early and later episode of malaria. **C** Cell-surface expression of indicated markers on CD39+ or CD39neg ZEB2+ memory CD4+ T cells of aged-matched protected (n = 8, Orange) versus susceptible participants (n = 8, Pink). **D** Co-expression of PD1 and LAG-3 on ZEB2+ memory CD4+ T cells of aged-matched protected (n = 8, Orange) versus susceptible participants (n = 8, Pink). Each symbol is one participant and P-values were calculated using two-sided Student's *t* tests with *p < 0.05, **p < 0.01. Pool of two independent experiments represented as individual data and means ± SEM (**C**, **D**).

---

cytolytic memory CD4+ T cells. The fact that these cells were more expanded in comparable aged-matched groups of individuals that either developed or did not develop clinical immunity during the length of our study, supports their contribution to protection, although more work is needed to confirm these findings.

Many studies have reported the existence of cytolytic CD4+ T cells (T$_{EM}$, T$_{EMRA}$), in large part during human viral infections[77]. These cells are prominent in viral infections that target HLA class II-expressing cells such as Cytomegalovirus, Epstein Bar Virus, Dengue, and Human Immunodeficiency Virus, and which include antigen-presenting cells (DC, B cells) and CD4+ T cells. In influenza and hepatitis C viral infections too, lung epithelial cells and hepatocytes upregulate cell-surface expression of HLA class II molecules, becoming targets for cytolytic CD4+ T cells. During viral infections (SIV, HIV, Influenza), these cytolytic CD4+ T cells may contribute to the control of viral replication through multiple effector mechanisms including direct cytolysis of infected cells. In malaria, T cell-dependent cytolysis has mostly been invoked as a mechanism to protect against liver-stage infection, through CD8+ T cell-mediating the rapid killing of MHC class I-expressing hepatocytes infected by *P. falciparum* sporozoites[78,79]. Yet, the liver stage of infection is time limited (up to a week), with a very low proportion of infected hepatocytes since mosquitoes only deliver a few hundred sporozoites into the host dermis, making it unlikely that memory CD8+ T cells developed in the course of endemic region *P. falciparum* infections could confer significant levels of protection. It has been shown, however, that irradiated sporozoite immunizations[25,76], and *Plasmodium* strains genetically engineered to arrest at late pre-erythrocytic stages[31], are able to promote high levels of parasite-specific immunity, likely through the production of high numbers of liver-resident parasite-specific memory CD8+ T cells[80]. Thus, upon repeated exposure to *P. falciparum* infection, it is conceivable that sufficiently high numbers of *Pf*-CSP specific memory CD4+ T cells with cytolytic features may accumulate systemically and perhaps reside in the liver of infected individuals, to help rapidly control sporozoite liver re-infections prior to the progression to the blood stage of infection occurs. In contrast to ZEB2neg counterparts, the ZEB2+ memory CD4+ T cells we describe express high levels of transcripts encoding for granulysin and multiple other effector molecules (granzymes, perforin, NKG7, IFNγ), as well as chemotactic receptors (CXCR3, CCR6) that may help their rapid migration or residency next to sites of injury where sporozoites first enter. *Plasmodium* sporozoites were indeed shown to glide along the liver sinusoids until they pause and traverse Kupffer cells or liver sinusoid endothelial cells to reach hepatocytes[79]. Kupffer cells express high levels of MHC class II and it is possible that upon recognition of parasite Ag, ZEB2+ CSP-specific memory CD4+ T cells arrest and release granulysin which could directly lyse paused sporozoites. Several studies suggested that CD8+ T cells can use such mechanism to lyse *Mycobacterium* bacteria and *P. vivax* parasites[81,82]. It is worth noting that ZEB2+ memory CD4+ T cells in age-matched protected compared to susceptible children, express lower levels of the "inflammatory site homing" chemotactic receptors CXCR3 and CCR6, and of the T cell exhaustion markers PD-1, LAG-3 and TIGIT. This seems consistent with the idea that these cells may be more restricted in their access to inflammatory sites (thus limiting systemic inflammation) while preserving higher functional features to clear the

parasites. Thus, it is tempting to speculate that they may represent important contributors of clinical immunity against *P. falciparum* malaria in vivo. Further investigations will be needed to define which antigenic peptides from CSP these cells recognize and if they may react to other sporozoite parasite Ag, and if and how they may confer immunity against liver-stage malaria. Such studies will need to overcome the present lack of reagents to track Ag-specific CD4+ T cells in malaria, due to the tremendous HLA class II diversity, parasite Ag variability and difficulty in characterizing strong *P. falciparum*-reactive epitopes across human populations.

More than 95% of the expanded clones sharing identical TCR clonotypes in protected participants during a malaria illness, expressed a robust T$_H$1 cytolytic effector program including transcripts encoding for multiple lytic effector proteins and the TFs ZEB2, T-BET, and RUNX3 (cluster 1). While we identified them among the CD45RO+ CD4+ T cells, these cells are reminiscent of CD45RA+ T$_{EM}$ (T$_{EMRA}$) cells that express a highly cytotoxic effector program and were reported to expand in viral infections (Dengue, Cytomegalovirus)[83,84]. The transcription factor (TF) ZEB2 was shown to drive the onset of terminally differentiated KLRG1hi effector CD8+ T cells in murine models of viral and bacterial infections, likely under the transcriptional control of T-BET[63,64]. Expression of T-BET and RUNX3 can turn off ThPOK and transcriptionally re-program mature CD4+ T$_H$ cells, the key T$_H$ cell master TF that suppresses the cytolytic program in CD4+ T cells[85]. These findings are consistent with the transcriptional signature of expanded clones in cluster 1. Interestingly, the "late" CD4+ T$_{EM}$ cells described in the *P. chabaudi (Pc)* murine model of chronic malaria infection[86], and were associated with improved parasite clearance, could represent murine equivalent of these cells. Another ~1.5% and ~0.5% of the expanded clones belonged to T$_H$1 cluster 2 and 3, respectively, which differentially express transcripts encoding granzyme K and ZEB2. Pseudotime analysis suggests that the expanded memory CD4+ T cell clones in clusters 2 and 3 represent intermediate, less differentiated stages, before they develop into robust cytolytic ZEB2+ memory cells. In particular, cluster 2 expanded clones express Eomes and IL-15 receptor beta chain transcripts, which regulates the expression of granzyme B in T cells. From the expanded clones, we also found that ~2% had a gene expression signature consistent with that of effector and regulatory T$_{R1}$ cells (cluster 4). From the pseudotime analysis, cluster 4 cells appear less differentiated than clusters 1-3, and on a distinct developmental trajectory, proliferating (Ki67+ and containing expanded clones) and with migratory features (CXCR6+) to or from infected tissues. In the mouse model of *Pc* malaria, highly proliferative CD4+ T$_H$1 intermediates exhibit either a T$_H$1 or a T$_{FH}$ fate[68,69], and some T$_H$1 cells also initiate the expression of genes associated with T$_{R1}$ cells (*Il10, Lag3, Ctla4*), consistent with our findings in the human *P. falciparum* malaria infection. While we did not detect any expanded clones sharing identical TCR clonotypes in the circulating memory T$_{FH}$ cells, it is possible that these cells are retained in lymphoid organs or infected livers and thus were not detected in the peripheral blood samples. The CyTOF analysis revealed a higher proportion of circulating memory CD4+ T$_{FH}$ cells in protected groups, which may also reflect the overall increase of this compartment in protected individuals. Further investigations are clearly needed to establish the origin of these cells and whether the CSP-specific ZEB2+ cytolytic memory

CD4[+] T cells expand in liver-draining lymph nodes before migrating to the peripheral blood and reach sporozoite-infected livers while T[FH] counterparts remain in draining lymph nodes and thus were not detected in the PBMC of *P. falciparum*-infected patients. The significantly higher proportion of classical monocytes (CD14[+]CD16[-]) in protected compared to susceptible participants that we and others[59,68] reported may influence the differentiation of the ZEB2[+] memory cells, as proposed in the mouse study[68].

This work includes a carefully defined classification of clinical immunity, and the application of advanced analytic methods including mass cytometry and single cell analysis to provide insights into protective immune responses during natural *P. falciparum* infection[87]. We provide one of the most comprehensive analyses of immune responses during mild malaria infection and expand upon prior focused studies to identify memory CD4[+] T cell subsets associated with clinical immunity[10,88–93]. We report the first single-cell analysis of CD4[+] T cell responses during human malaria infections for a granular analysis that defines subsets, characterize their potential functions and track expanded clones sharing identical TCR clonotypes and their antigenic specificity. This approach has recently been conducted in the mouse model of malaria[68,69] and expands upon prior whole blood transcriptional analysis to provide insights into key adaptive cellular subsets and potential functions during the human infection[87]. In summary, our study has identified humoral and cell-mediated signatures associated with clinical immunity, and the identification of a clonally expanded subset of ZEB2[+] memory CD4[+] T cells with a robust cytolytic phenotype that recognize the pre-erythrocytic stage CSP Ag. Signatures of immunity may vary by malaria endemicity, host and parasite factors and malaria control interventions. Similar analyses in other regions and in RTS,S-vaccinated individuals could provide further insights into the importance of these cells in malaria immunity[94–96].

## Methods
### Study design
To identify immune correlates of clinical immunity in a high transmission malaria endemic region we recruited study participants who presented with mild malaria into a prospective 18-month longitudinal study to enumerate malaria reinfections over time at the Mfera Health Center in Chikwawa District under the auspices of the Malawi International Center of Excellence for Malaria Research[97]. The Chikwawa District of Malawi is a wet, low-lying rural district in the Shire Valley with high, year-round malaria transmission[98].

Enrollment criteria required an episode of mild malaria which was defined by the WHO as presenting symptoms compatible with mild malaria and without signs of severe malaria. Participants had positive rapid diagnostic test for malaria (SD BIOLINE, Malaria Ag *P. f.*, Abbott) at the field site, which was later confirmed positive by microscopy. Patients with chronic medical conditions including a history of HIV infection were excluded from enrollment. To capture variation in naturally acquired clinical immunity in a highly malaria-endemic region, 40 participants in each of 3 age groups (1–5, 6–12, and 13–50 years) were enrolled. After the enrollment visit, study participants underwent monthly clinic study visits and interim visits during illness to evaluate symptoms of malaria, record temperature, collect a blood smear and a blood drop onto a FTA classic cards (Whatman, NJ) for PCR detection of *Pf* [97]. Recurrent clinical malaria was defined as an episode with clinical symptoms consistent with malaria and a positive malaria blood smear (>2500 parasites/μL)[99]. Parasitemia was determined by microscopy and was recorded as the geographic mean value of two independent readers. Whole blood samples were collected in ethylenediaminetetraacetic acid (EDTA) coated tubes at enrollment, during each subsequent episode of clinical malaria and day 30 post-infection. Within 6 h of collection, the blood sample was centrifuged to collect plasma aliquots and isolate PBMCs using ACK lysis buffer (Crystalgen) with standard methods[100]. PBMC aliquots were stored in

5% DMSO and 95% FBS (Fisher Scientific) in liquid nitrogen. PBMC and plasma samples were shipped to Albert Einstein College of Medicine in New York, in liquid nitrogen gas and stored in liquid nitrogen for subsequent immunophenotyping and protein array analysis. Filter papers were collected during each episode of clinical malaria and routinely at each monthly visit for parasite genotyping[97]. An infection was counted if the malaria genotype by sequence analysis was distinct from the prior infection or occurred at least two weeks after a prior infection in each participant (including episodes of non-clinical malaria and submicroscopic infections). Participants who were diagnosed with clinical malaria were treated with 3 days of artemether/lumefantrine per Malawi Ministry of Health National Treatment Protocol. Participants were considered lost to follow-up if they did not present for > 3 consecutive monthly study clinic appointments.

Informed written consent was obtained prior to study enrollment. Institutional Review Board approvals were obtained from the Albert Einstein College of Medicine, Michigan State University, the University of Maryland, and the University of Malawi College of Medicine Research and Ethics Committee.

### Detection of sickle cell polymorphism
Sickle cell polymorphism was an exclusion criterion for further immune analysis. To determine sickle cell status, blood was spotted onto FTA classic cards (Whatman, NJ) from the enrollment whole blood sample in EDTA tubes. Whole DNA was extracted using QIAamp DNA mini kit (Qiagen, Hilden Germany) following the manufacturer's instructions. Following whole DNA extraction, the sickle cell allele status was determined by restriction fragment length polymorphism (RFLP) on PCR amplified DNA and gel electrophoresis following previously published protocols[101]. Genomic DNA (gDNA) from a patient homozygous for the sickle cell trait was used as a positive control.

### Identification of participants gaining or not gaining clinical immunity
To identify participants who gained immunity over the 18-month study, we estimated participant-specific rates of recurrent clinical malaria per additional exposure event by fitting a mixed effects regression model with the total number of recurrent clinical malaria as the binary outcome, the number of exposure events as the main predictor, and included a random intercept and a random slope (i.e. a random coefficient to the number of exposure events). Only participants who had at least two clinical malaria episodes during the study period were included. Recurrent clinical malaria was defined by a temperature of ≥37.5 °C with a *P. falciparum* parasitemia density ≥2500 parasites/μL. Based on these models, we estimated conditional average trends and participant-specific trends. The participants were then ranked by predicted value of participant-specific rates of clinical malaria (i.e. conditional modes of the random slope)[102].

### Immunophenotyping by cytometry by time of flight (CyTOF) and spectral flow-cytometry
Frozen vials of PBMCs were placed in a 37 °C water bath and gently agitated until ~90% was thawed. The vials were transferred to a tissue culture hood and 1mL of sterile warmed media (RPMI/20%FBS) was added to the PBMC for 5 min before transferring drop-wise to a 10mL volume of warmed media (RPMI/20%FBS) containing 200U/ml of DNase I (Roche, Switzerland). After 10 min at room temperature, the PBMC were centrifuged at 300xg, at room temperature for 5 min. The pellets were washed twice with room temperature media (RPMI/20% FBS) before transferring to 96 well plates for staining with antibodies for CyTOF or spectral flow cytometry.

*For mass cytometry*, PBMC samples were labeled with cisplatin, stained with heavy-metal conjugated Ab listed in Supplementary Data 4, and barcoded following the manufacturer's protocol (Fluidigm, US) with minor modifications. Ab not commercially available as

conjugated with specific heavy metal ions were conjugated using the ready-to-label antibody format and Maxpar Antibody Labeling kit (Fluidigm, US). Briefly, cells were stained with extracellular Ab in Maxpar staining buffer for 30 min on ice followed by the addition of cisplatin ($5\mu M$ in PBS) for 2 min at room temperature. Cells were washed twice in Maxpar staining buffer followed by fixation and permeabilization using the eBioscience FoxP3/transcription factor staining kit (Thermofisher Scientific) for intracellular staining and barcode labeling. Lastly, barcoded samples were combined for intercalation using Cytofix/Cytoperm buffer (BD) and 2% PFA/Ir solution for 30 min at room temperature. This was followed by a wash and resuspension of cells in MaxPar cell staining buffer with 125nM of Ir and storage at 4 °C before running on the Helios CyTOF instrument (Fluidigm) at the Icahn School of Medicine Mount Sinai, Human Monitoring Core (New York, NY). The data was analyzed on FlowJo LLC software (v10.7.1, BD, Ashland OR).

*For spectral flow cytometry*, participant's PBMCs were first stained with Live/dead Fixable Aqua (Thermofisher Scientific) for 30 min at room temperature. Cells were washed and incubated with human Fc block (BD) for 15 min on ice, followed by staining for extracellular markers with fluorochrome-conjugated antibodies (Supplementary Data 4) in FACS buffer (PBS, 1% FCS, 0.02% Sodium Azide, 2mM EDTA) and 50μl of BD Brilliant stain buffer (BD) for 30 min on ice. Cells were washed twice and then fixed and permeabilized with the eBioscience FoxP3/transcription factor staining kit (Thermofisher Scientific). Cells were resuspended in blocking permeabilization buffer (1% mouse serum, 100U/ml heparin, 0.2% BSA) for 20 min at room temperature before the addition of fluorochrome-conjugated antibodies against intracellular markers (Supplementary Data 4) on ice for 30 min. Cells were washed and resuspended in FACS buffer and stored at 4 °C before acquisition on the 5 laser Cytek Aurora instrument (Cytek Biosciences). The data was analyzed on FlowJo LLC software (v10.7.1, BD, Ashland OR).

### Cell sorting for single-cell RNA-sequencing

PBMCs isolated during malaria infection were thawed as described above and stained with Live/dead Fixable Aqua (Thermofisher Scientific) for 30 min at room temperature. Cells were washed and incubated with antibodies against CD3, CD4, CD19, IgD, IgM, CD45RO, CD27, and CXCR3 on ice for 30 min in MACS buffer (PBS, 2% BSA, 2mM EDTA). Cells were washed, resuspended in a solution of 50% MACS buffer and 50% FBS, and run on the BD Aria III with the 100μm nozzle. Up to 19,000 live naïve CD4$^+$ T cells (CD19$^-$ CD3$^+$ CD4$^+$ CD45RO$^-$ CD27$^+$) or memory CD4$^+$ T cells (CD19$^-$ CD3$^+$ CD4$^+$ CD45RO$^+$) were sorted into the solution of 50% MACS buffer and 50% FBS. Sorted cells were applied to the 10x Genomics single-cell RNA-seq platform for cell barcoding, cDNA synthesis and 5' gene expression library and V(D)J library preparation following the manufacturer's protocols. 5' gene expression and V(D)J libraries were sequenced on the Hi-seq Illumina platform (2 x 150bp paired-end, Genewiz).

### Single-cell RNA-seq analysis and differential expression analysis

Unique Molecular Index (UMI) Count Matrices for gene expression were generated using the CellRanger count (Feature Barcode) pipeline (v2.1.1). Reads were aligned on the GRCh38-3.0.0 transcriptome reference (10x Genomics). Chemistry was Single Cell 5' R2-only. Filtering for low-quality cells according to the number of RNA, genes detected, and percentage of mitochondrial RNA was performed. Then, gene expression matrix was regressed for cellular sequencing depth and mitochondrial percentage using linear modeling as implemented in Seurat ScaleData function (Seurat v3.2.2).

Datasets were integrated within each condition, naïve and memory representing a total of 6,664 and 7,934 cells respectively using the Seurat Canonical Correlation Analysis (CCA) and graph-based integration tool on the 3,000 most expressed genes across datasets to correct for batch effect (Supplementary Code 1). The 30 first dimensions of the PCA of the batch effect corrected matrix was used to generate the Shared Nearest-neighbor (SNN) graph and the UMAP. Graph-based clustering using Louvain algorithm with a resolution parameter of 0.46 on the FindCluster function was used to cluster cells. Each cluster was annotated using cell type-specific markers. Markers for each cluster were identified using FindAllMarkers function with default parameter. Genes were then ranked based on their expression fold change, the difference of detection of this gene in the cluster versus all other clusters and the specificity for the cluster, and top cluster-specific genes were compared with published cell type-specific genes to annotate the cluster. Up and down-regulated genes for each cluster were identified using likelihood-ratio test for single-cell gene expression comparing the cluster of interest with all other clusters. P-values for reported genes were <0.01.

### Single-cell RNA-seq TCR analysis

Raw data from the TCR libraries were first processed with the Cell-Ranger VDJ pipeline (ChemistrySingle Cell V(D)J). We used V(D)J Reference GRCh38-alts-ensembl and Cell Ranger Version 2.1.1. The output TCR sequences in each cell were integrated with their expression data using the shared cell barcodes within each sample. Downstream analyses were applied in R/ Seurat package (v3.2.2). We use clonotype frequency distribution to define clonally expanded cells. The threshold used based on the distribution tail was a clonotype frequency greater than 10. Differential expression analysis between expanded versus non-expanded was performed using likelihood-ratio test for single-cell gene expression. P-values for reported genes were <0.01.

### Pseudotime analysis

Differentiation trajectory analyses were conducted with monocle[71] (monocle3). Preprocessed Seurat object were imported using *importCDS* function from the monocle R package. Monocle's *orderCells* function was used to arrange cells along a pseudo-time axis to indicate their position in a differentiation continuum. Monocle generates for each cell a pseudotime value with respect to predefined cell of origins (roots). Here the $T_{CM}$ cells were used as roots. We specify the root of the trajectory programmatically, as recommended, by first grouping the cells according to which trajectory graph node they are nearest to. Then, calculate what fraction of the cells at each node come from the earliest time point followed by selecting the node that is most heavily occupied by early cells and returns that as the root.

### Parasite culture and merozoite opsonization assays

*Pf* 3D7 parasites (MR4) were cultured in vitro in parasite media (RPMI, 0.6% HEPES, 0.05% gentamicin sulfate, 0.005% hypoxanthine, 0.2% sodium bicarbonate, 0.5% Albumax II, 0.0001% 0.5N NaOH, pH 7.4) with 5% hematocrit in plugged flasks with 1% $O_2$, 5% $CO_2$, 94% $N_2$ gas mixture at 37 °C. Cultures were maintained at 2–4% parasitemia. Merozoite opsonization assay was adapted from previously published protocols[103,104]. Briefly, *Pf* 3D7 cultures at ~2-6% parasitemia were enriched for late stages (trophozoites and schizonts) after diluting the culture by 50% in parasite media and passing it over magnetic LS columns (Miltenyi). Late-stage enriched *Pf* 3D7 cultures were incubated for 12 h in 10μM E64 (Epoxysuccinyl-L-leucylamido 4-guanidino butane) supplemented parasite media at 37 °C with 1% $O_2$, 5% $CO_2$, 94% $N_2$ gas mixture. Late-stage enriched *Pf* 3D7 cultures were then passed through a 1.2μm filter to release merozoites and then passed over a magnetic MS column Fito (Miltenyi) to recover merozoites without hemozoin. Isolated hemozoin-free merozoites were labeled by incubation with 0.5μM YOYO-1 Iodide dye (Thermo Fisher) for 30 min at room temperature. Labeled merozoites were washed and centrifuged at 3700 x g for 10 min before resuspension in RPMI media. Absolute numbers of labeled merozoites were determined by flow

cytometry and CountBright™ Absolute Counting Beads (Thermo Fisher). Labeled merozoites were incubated with US control plasma or participant plasma (1:10 dilution or 1:100 dilution) or no plasma for 1 h at 37 °C. Opsonization assay was established with a ratio of 1 THP-1 cell to 10 labeled merozoites pre-incubated with participant plasma (~2400 THP1 cells: 24,000 labeled merozoites) in FBS-blocked 96 well U-bottom plates for 10 min at 37 °C. Opsonization was terminated by placing cells on ice and washed with cold FACS buffer and centrifuged at 300g for 10 min at 4 °C. Cells were fixed in 2% PFA for 10 min on ice. Cells were washed, resuspended in FACS buffer and stored at 4 °C before acquisition on the BD Aria III. The percentage of THP-1 cells positive for YOYO-1 DNA staining was reported.

### Activation induced marker (AIM) assay

Cryopreserved PBMCs were thawed, washed, suspended in complete medium (RPMI1640 Glutamax, supplemented with β-mercaptoethanol, pyruvate, pen/strep, and 10%FBS) and cultured at 2-3 x $10^6$ cells/well in 96-well U-bottom plates with parasite Ag or control peptides. Recombinant purified Ag from CSP (amino acids Gly86 to Ser410 of the full-length CSP), AMA-1, MSP-1, EBA-175 or control *E. coli* derived Exo Protein A (EPA) and *P. pastoris (Pp)* derived Pfs25M (each at 5μg/mL) were incubated with PBMCs for 24 h at 37ºC in presence of anti-CD28 (clone L293) and anti-CD49d (clone L25) mAb at 1 μg/mL (BD Biosciences). Malaria Ag were either expressed in *Pp* (AMA-1 FVO, CSP) or *E. coli* (MSP-1$_{42}$ FVO). Control Ag were also expressed and purified from *Pp* (Pfs25M) or E. coli (EPA)[105]. These Ag were provided by Dr. Patrick Duffy, NIAID, NIH. Recombinant EBA-175 RII was expressed in *P. pastoris* and obtained through BEI Resources, NIAID, NIH (MRA-1162), and contributed by Annie Mo. After 24 hrs, PBMCs were washed with FACS buffer and incubated with anti-Fc blocking Ab for 15 min at 4 °C prior to surface staining with the mAb against CD3-eFluor450 (UCHT1, 1/20), CD4-APC (SK3, 1/10), CD25-Per-CP (M-A251, 1/20), CD45RO-FITC (UCHL1, 1/10), CD134-Biotin (ACT-35, 1/20), 41-BB-PE-Cy7 (4B4-1, 1/20) for 30 min at 4 °C, followed by incubation with streptavidin PE for 20 min. Cells were then fixed and permeabilized with the FOXP3/Transcription Factor Staining Buffer Set following the manufacturer's instructions (eBioscience) and stained with anti-ZEB2-AF700 (1/60) for 20 min at 4° C in the dark. Stained cells were collected on BD Aria III.

### Analysis of *P. falciparum* antigen microarray probed with participant plasma

*Plasmodium* arrays were constructed as previously described[54]. The Pf900/Pf250 array used in the present study, comprising 826 unique features corresponding to 676 unique *Pf* genes, was a down-selected array based on results from previous microarray field studies[39,54,55,106]. Each microarray chip contained multiple negative in vitro transcription/translation control spots that lack plasmid template and serially diluted human immunoglobulin (Ig) G and anti-IgG control spots.

Plasma samples diluted to 1:100 in Protein Array Blocking Buffer (Whatman) were preincubated in 10% of 1mg/ml *Escherichia coli* lysate, and microarrays were probed with plasma samples by incubation overnight at 4°C. The slides were washed 5 times in Tris buffer (pH 7.6) and incubated in Qdot 800-conjugated goat anti-human IgG diluted at 1:100 (Grace Bio-Labs, Inc., Ref # RD480080). The slides were washed 5 times in Tris buffer containing 0.05% Tween 20, followed by a final wash with water. Air-dried slides were scanned at 500ms and quantified using an ArrayCam 400-S Microarray Imaging System (Grace Bio-Labs, Incco transformed. A log2 (FOC) value of 0.0 means that the intensity is not different from the background intensity and a value of 1.0 indicates a doubling with respect to the background. A threshold of a $log_2$ (FOC) of 1 was used to define seropositivity. Ag were considered reactive when mean log2 (FOC) among any of the groups is greater than 1.0.

### Statistics and reproducibility

We conducted an observational longitudinal study of a cohort of Malawian children and adults to detect malaria infections using active and passive surveillance over 18 months. To enroll sufficient numbers of malaria-susceptible and protected participants and to allow subanalyses by sex, we conducted sample size calculations based on prior malaria epidemiologic data from Malawi. Categorical data such as self-reported sex was compared using the Chi-squared test. Numerical data such as age, number of malaria infections and cellular subsets were analyzed using Student's, Welch's *t* test, Mann–Whitney test or two way ANOVA test as appropriate. For the analysis of the protein array data, Differential analyses of the log2-transfomed data were performed using a Bayes-regularized *t* test for protein arrays. Differences were considered significant with Benjamini–Hochberg corrected *p*-values < 0.05. The antibody profile breadth was defined as the number of seropositive Ag (log2 (FOC) > 1) recognized by an individual or for a population using the mean signal intensity for the population. Mann–Whitney *p* values were calculated by comparing the distribution of signals among different groups. Analysis was performed using the R statistical environment (http://www.r-project.org). Graphs were produced in R, Graphpad and Excel software programs.

### Reporting summary

Further information on research design is available in the Nature Portfolio Reporting Summary linked to this article.

## Data availability

The accession number for the single cell RNA-seq and TCR-seq data reported in this paper is GEO:GSE182536. All other data is available in the main text or the supplementary information and source data files. Source data are provided with this paper.

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

## Acknowledgements

The authors thank The Albert Einstein FACS and genomic core facilities (Dave Reynolds), and the Mount Sinai School of Medicine Mass Cytometry core (Adeeb Rahman). This work was funded by the National Institute of Health Grants (NIH/NIAID) AI138552 and 3U19 AI089683-03S1 to G.L. and J.P.D., R21AI141367 and R01AI164864 to J.P.D., 4U19AI089683 to T.T., T32 AI007291 to J.S., the Hirschl Caulier Award and the Sylvia and Robert Olnick Faculty Scholar in Cancer Research to G.L. and Grant P30CA013330 and S10OD026833 to the Flow Cytometry Core Cancer Center. R.F. was supported by the NIH BETTR IRACDA training grant K12GM102779. We thank biorender.com for Fig. 7B.

## Author contributions

R.F., J.P.D. and G.L. designed and interpreted most experiments and figures and wrote the paper. M.P. did all revision experiments (AIM assays), data reanalyzes and manuscript editing. R.F. and G.L. with J.Z. help and input, designed and validated CyTOF and spectral flow cytometry panels. K.S., Andy Bauleni, Andrea Buchwald, S.G., M.L., T.T. and J.P.D. contributed to the clinical study design, study management and analysis. P.K. conducted the sickle cell analysis. F.D. with R.F., J.P.D. and G.L. analyzed, interpretated all transcriptomic data and contributed to related figures. L.L., R.F. and P.F. performed protein microarray probing and analysis and contributed to the related figures and Data. J.S. and S.C.Z. contributed to manuscript preparation. R.S.K. and J.S. contributed to the biostatistical analysis. R.S.K. was the consulting statistician.

## Competing interests

The authors declare no competing interests.

## Additional information

[1]Department of Microbiology and Immunology, Albert Einstein College of Medicine, Bronx, New York 10461, USA. [2]Department of Medicine, Albert Einstein College of Medicine, Bronx, New York 10461, USA. [3]Department of Epidemiology and Population Health, Albert Einstein College of Medicine, Bronx, New York 10461, USA. [4]Université de Paris, AP-HP, Hôpital Saint-Louis, Laboratoire d'Immunologie et Histocompatiblité, INSERM UMR976, 75010 Paris, France. [5]Department of Physiology and Biophysics, School of Medicine, University of California, Irvine, CA 92697, USA. [6]Malaria Alert Centre, Kamuzu University of Health Sciences, Blantyre, Malawi. [7]Blantyre Malaria Project, Kamuzu University of Health Sciences, Blantyre, Malawi. [8]Center for Vaccine Development and Global Health, University of Maryland School of Medicine, Baltimore, MD 21201, USA. [9]Department of Osteopathic Medical Specialties, Michigan State University, East Lansing, MI 48824, USA. [10]Department of Genetics, Albert Einstein College of Medicine, Bronx, New York 10461, USA. [11]FD: Precision Oncology, Sanofi, Vitry sur Seine, France. [12]Present address: RF: BioNTech US, 40 Erie Street, Cambridge, MA 02139, USA. [13]Present address: Division of Infectious Diseases, Johns Hopkins University School of Medicine, Baltimore, MD 21205, USA. [14]These authors contributed equally: Raquel Furtado, Mahinder Paul. ✉e-mail: johanna.daily@einsteinmed.edu; gregoire.lauvau@einsteinmed.edu

