## [Peer Review File · Nature Communications]

Cytolytic circumsporozoite-specific memory CD4⁺ T cell clones are expanded during *Plasmodium falciparum* infectionREVIEWER COMMENTS

Reviewer #1 (Remarks to the Author):

Furtado and colleagues provide data from a study of 97 individual from Malawi selected on presentation with mild malaria confirmed by blood smear. This cohort was monitored monthly for 18 months as well as upon hospital presentation with re-infection. Re-infection was confirmed by the presences of symptoms of malaria and >2,500 parasites/ul blood. The study focused on 3 sub-groups based on age (<13 or older) and disease susceptibility. The clinically susceptible cohort (all <13) were defined as having 3 or more reinfections in 18 months, whereas the protected cohorts were defined as having 1 or less infections in the following 18 months. Protected individuals were separated into two cohorts, <13 and older to enable age-matched comparisons with the susceptible cohort. The authors showed that the protected older cohort had broader plasma IgG reactivity and higher titers against Pf Ags, and increased functional merozoite-specific IgG. Classical monocytes, circulating Tfh and memory B cells were increased significantly in both protected cohorts, whereas memory CD4 T cells were solely increased in older protected group. These data suggest that increase classical monocytes, Tfh and memory B cells in the blood are associated with protection and support cohort divisions.

The authors then went on to examine transcriptomes and TCR gene expression in naïve and effector/memory T cells from 3 protected individuals undergoing infection (not mentioned which age cohort). They showed that memory T cells could be clustered into 8 distinct groups, largely consisting of Th1, Th2, Tfh, Tcm and regulatory T cells. Analysis if TCR clonotypes revealed that most expanded clones belonged to what appears to be the most differentiated Th1 type of memory CD4 T cells that expressed a robust cytolytic program. These cells were implicated in malaria immunity but no transcriptional data was provided to show that these cells expanded upon malaria infection. The authors then showed that the genes for Zeb2 and granzyme B were preferentially expressed by the expanded clones, suggesting the protein products may be useful surrogate markers for malaria-specific cells that have clonally expanded. They then showed that Zeb2+ but not granzyme B+ cells were expanded during acute infection of both younger cohorts, but not in older protected individual. The authors propose that these findings suggest that this type of CD4 T cell population (Zeb2+) may be important for protection of younger cohorts, but don't provide any direct evidence for protection.

While the study represents a nice assessment of many aspects of immunity in humans undergoing endemic infections with Plasmodium species, there are no clear links between observed phenotypes and disease or disease outcome. The first part of the paper makes some previously reported associations to support cohort divisions but the later parts of the paper are really only descriptive. There is a lot of speculation in the discussion but it is really only speculation that is loosely supported by the findings. I have highlighted some major and minor issues below.

Major issues:

Major issue 1. The authors draw the conclusion "The broader plasm IgG reactivity and titers against Pf Ags, together with the increased functional Pf-merozoite specific IgG responses in the protected compared to the susceptible participants, further verifies the clinical immunity classifications for the in depth studies of their immune cells presented below." But, these differences are generally only seen between the older protected group and the younger unprotected group, and not between the two younger age-matched groups. Thus, this softens justification for the proposed comparison and means that any significant differences could be correlates with age rather than immune status.

Example: Number of Ags recognised and amount of IgG were higher in protected older group, but not different between protected and unprotected age-matched younger groups.

Related to this issue, while the data support the conclusion as written, the following sentence is misleading as it hides the fact that there is no significant difference between age-matched protected and unprotected groups: "Approximately 50% of THP-1 cells were YOYO-1+ when co-cultured with merozoites sensitized by preincubation with clinically immune participants plasma of all ages. In comparison, only 35% of THP-1 cells stained positive for YOYO-1 after co-culture with merozoites preincubated with plasma from the susceptible group"

Major issue 2. Evidence that Th1 cytolytic phenotype cells make up the majority of expanded clones in the blood is interpreted to mean that this type of effector is preferentially induced during malaria and may be important for protection (- no data on the latter though). However, this observation could relate to a preference for this phenotype of clones to largely reside in the blood, compared to other malaria-specific populations. If, for example, various subsets of memory T cells were expanded in response to malaria after infection: some that go into tissues to form tissue-resident populations, some that go into the lymph nodes and spleen (like Tcm) and some that form a vascular-resident population, then the latter may be over-represented in expanded clones in the blood. A paper by Bugget et al. suggests that this may be the case for CD8 T cells in humans (Cytolytic CD8 T cells confined to intravascular circulation. Buggert et al. Cell 183: 1946-1961, 2020). This type of outcome might mean that many different subtypes of CD4 T cells are formed in response to malaria but we preferentially only see the cytolytic CD4s expanded in the blood. This would not mean they are a critical protective subset, but simply that they are more visible when sampling blood.

Major issue 3. The authors finally conclude that because a large proportion of expanded clones express Zeb2 and they find that Zeb2+ cells are found to be expanded in the blood of the two younger cohorts upon infection, these cells may be protective. However, their data shows no difference between protected and susceptible age-matched cohorts. This seems to argue that they are not the key population required for protection.

Minor

1. No y axis label on Figure 1E
2. Reactivity to LSA3 is implicated to be to liver-stage as wording implies it is only a liver-stage Ag but based on Malaria Cell Atlas, it appears to be expressed also during the blood stage. Might need to reword reference to this to allow reader to consider that this immunity may have been generated to blood stage Ag exposure.
3. Figure 3B. I am not sure that it is correct to do t test without correcting for multiple comparisons. Also, I think that Myeloid DC, pDC and NK cells need to be on a different scale to show variation as these subsets are a very small proportion of the total cells.
4. Statement line 266 that class-switched B cells are increased in both groups is not supported by the data (Figure 1C). This figure only shows that memory B cells are increased in both protected groups.
5. Figure S4B, E. I think it would be clearer if the individual's data were shown on separate UMAP plots next to each other to enable comparison of cluster make up. It is not clear to me that participant 2 has cells in the Tfh cluster in S4B for example, and S4E is so full of colors it is hard to decipher.
6. What age were the 3 protected individuals used for transcriptomics?
7. I am not clear how relative expression is calculated in Figure S5B. Is there a threshold that is set as positive then the clones are assessed for how many individual cells in the clone pass this threshold?
8. Figure 7A. What is y axis denominator? Is it white blood cells or CD4 T cells or CD4 memory T cells? Clearly outline how this relates to the proportion of Th1 cluster 1 cells you found earlier. Do these numbers match at all? Why don't you have any CD4-negative T cells that are Zeb2+. Should this also come up on CD8 effectors?

Reviewer #2 (Remarks to the Author):

This is a really interesting and ambitious look at the combinatorial factors leading to disease resistance in *P. falciparum*. They show that patients in their study area under 13 are still largely susceptible, but can identify some individuals with fewer reinfections over the study period who have less anemia, lower parasitemia, fewer episodes of fever, and higher quality opsonizing antibodies, but not more Pf antigens recognized (for younger immunes). That is just up to Figure 2! They used a combination of multidimensional assays (CyTOF, spectral flow cytometry and single cell transcriptomics) to identify immune correlates of immunity. It is interesting (and potentially useful to note in the abstract) that there were increases in Tfh, Isotype switched B cells and classical monocytes seen in protected subjects.

The population that stood out of these analyses was a little-studied highly differentiated cytolytic CD4 T cell subset. While this is interesting, and MHCI recognition has been shown to kill infected *P. vivax* infected immature red blood cells, it is not clear how MHCII recognition can lead to beneficial killing of the parasite, though killing of infected Kupfer cells, or phagocytes or antigen presenting cells could have outsized effects on immunoregulation as well. It seems likely that the killing function is incidental to the highly differentiated Th1 phenotype described. The discussion could be streamlined a bit potentially by focusing on highly differentiated Th1-like memory cells, called Teff Late, as described in Opata et al, 2018; which are protective and also can be generated by CD27-Teff Late (Opata 2015) that protect and express several Granzyme genes, as seen in the transcriptomics GEO GSE89555. Perhaps Th1-like mouse malaria papers (Zander/Butler, Wikenheiser/Stumhofer et al) can also shed some light. Alternatively, are there individual markers that are significantly differently expressed in T cells from protected participants that could be used to distinguish these children in future studies? Another idea is to look for FcR, which have been seen upregulated on T cells in malaria, NK cells with FcR can kill RBC with antibody using granzymes.

The paper does not seem to come full circle in identifying T cell features that are different between protected and susceptible children, but the groundwork has been nicely laid for future studies to determine potentially protective attributes discovered here. Were expanded memory T cell subsets only seen in protected participants, but not susceptible? Are there inhibitory co-stimulatory molecules that are significantly changed in protected but not susceptible participants (Zeb2 and GzmB? Or CX3CR1/CXCR3, CXCR5, CCR6!)? If so, list in abstract.

Minor suggestions

-Show controls in S2C in Figure 2C

-Perhaps its confused by the lack of specificity in the T cell analysis, but GATA3+ Th2 cells and Tcf7+ Tcm seem unexpected in malaria. Do Th2 correlate with worm exposure (if you have this clinical info)? Do these cells also express IL-4, 5 or 13, upregulated STAT6? ST2? If not, perhaps identifying the population as GATA3+?

Not sure we can call Tcf7 a master regulator of Tcm yet. Surprising that quiescent Tcm would be a large subset in malaria too, particularly in a region of repeated exposure. What does the metabolism transcriptome suggest? The cell cycle genes?

--results section for costimulatory and inhibitory molecules (lines 394-401) is not clear, do young protected participants have more or less inhibitory or costimulatory molecules on their expanded clones or total Tmem?

Reviewer #3 (Remarks to the Author):

Furtado et al have performed a comprehensive longitudinal study of participants living in a malaria endemic area of Malawi. Participants were classified as either susceptible or protected from *Plasmodium falciparum* (Pf) infection based on the number of recurrent clinical infections over the 18-month study period. Their findings demonstrate age related differences in susceptibility to *Plasmodium falciparum* infection and clinical immunity was also dependent on these age differences. They performed detailed analysis on plasma and peripheral blood mononuclear cells (PBMCs). Compared to susceptible participants, they found higher plasma IgG titres and broader IgG to Pf antigens in protected participants as well as greater antibody dependent Pf opsinisation. High dimensional flow cytometry was performed on cryopreserved PBMCs for immunophenotyping and single cell transcriptomics was performed on 3 protected participants. The study shows some interesting results, in particular an expansion of memory CD4 T cell clones with cytolytic characteristics.

Specific comments are found below:

Results

- Line 140: the authors state they enrolled equal numbers of children, adolescents and adult participants but the data in figure S1A is confusing as it shows a table with the participant age break down as 1-5 years, 6-12 years and 13-50 years. Generally, the definition for a child would be <12 years old, adolescent 13-18 years old and adult >18 years old. Can the authors explain the reason for the breakdown of age brackets in the table shown in S1A?
- It becomes apparent in the next section of the manuscript (line 153) that new definitions need to be set for each malaria endemic region and this then sets new age brackets and 3 populations that are further studied in detail. Perhaps for clarity it would be better to include a table of the study characteristics (as those listed in S1A) of the 3 groups – Susceptible <13yrs, Protected <13yrs and Protected >13years.
- Line 213: What is the cut-off for the YOYO-1 fluorescent dye/assay? Has this been defined?
- Line 234: Table S4 has been labelled Table S5
- Line 236: CD62L expression is generally best detect on fresh blood and difficult to detect on frozen PBMCs, it is unclear from the gating strategies presented whether CD62L was even used for gating – please clarify,
- Line 245 – 253: the authors discuss significantly higher frequencies, however no values are included, in my opinion the authors should report the medians and p-values in the text.
- Line 256: It must be noted that the TFH found in the periphery are not bona fide TFH cells but circulating TFH - also it is unclear if these have been gated on total CD4 T cells or memory CD4 T cells, this could lead to a difference in results between groups – please clarify.
- Gating of Treg cells is also unclear if based on total CD4 or memory CD4 – please clarify. Also gating of Treg cells is more reliable if CD25 and CD127 are included see Sedikki et al <https://www.ncbi.nlm.nih.gov/pmc/articles/PMC2118333/pdf/jem2031693.pdf>
- Line 287: Assume this is the correct Table S5 as per uploaded version.
- Line 316: Does the TFH cluster express PD-1 and CXCR5?
- Line 368: Have the authors checked to see if these clones are specific for Pf?
- Line 385: Same comment as above for CD62L
- Line 387: Are the authors suggesting the Zeb2+ CD4 T cells that upregulate CD39 are specific for Pf? In combination with CD25 and CD134, CD39+ CD4 T cells have been shown to be antigen-specific Tregs (<https://pubmed.ncbi.nlm.nih.gov/24752698/>).
- Line 394-401: No reference to the figures.
- Line 405: There is no indication that these cells are antigen specific, would be a nice addition to the current data if possible.
- Line 475: would be great to show that these cells actually have cytotoxic potential if possible, perhaps in a killing assay?
- Line 601, 618, 624: Assume Table S5 is actually Table S4 (as per upload)

Methods:

- Unclear if experiments were performed on PBMCs from different time points, please clarify
- ### Figures
- Figure 1C: interesting there seems to be 2 groups in the susceptible population with regards to temperature – can the authors comment?
 - Figure 1E: the label for the left y-axis is missing
 - Figure 2A: x-axis missing
 - Figure 4B: Authors use CD45RA to gate for naïve CD8 T cells, can the authors be sure these are not CD8 revertants or Temra?
 - Figure 7A: x-axis missing from graph on right
 - Figure 7B: bit confusing what the message is here are the Zeb2+ protected data from all ages?
 - Figure 7C: is the key missing for susceptible and protected?
 - Figure S6B: x-axis missing from graph on right

Reviewer #4 (Remarks to the Author):

This assessment is only in regard to the systems biology component of this work.

Results

The bioinformatics analysis is sound, although the methods section lack key details.

The scRNAseq +TCR was performed only on 3 protected individual, which somehow contrast with the main narrative of the manuscript.

It would be more sound to consider more scRNAseq data by splitting <13 and >13 , or compare susceptible versus protected. The bioinformatic analysis should likely reveal clear signatures, including in the cytotoxic CD4

How was the comparison between clonally expanded and non expanded cells performed? What test? Zeb2 was identified but was this significant ?

Could naive T cells be used as root? It would be important to see if the same clone can be shared with the naive cells

The method section should include more details.

- scRNAseq bioinformatics method is not fully explained.

How is the trajectory analysis performed? What is the rationale behind trajectory analysis? How is the route chosen? How is the clonal analysis performed? What statistical tests were performed and what cut off were used.

- Batch effect and other QC steps, this reviewer did not find any details on these QC steps of the scRNAseq.

Were data batch corrected? Were cells excluded and with what details?

Point by point reply to reviewers's comments:

R#1:

Furtado and colleagues provide data from a study of 97 individual from Malawi selected on presentation with mild malaria confirmed by blood smear. This cohort was monitored monthly for 18 months as well as upon hospital presentation with re-infection. Re-infection was confirmed by the presences of symptoms of malaria and >2,500 parasites/ul blood. The study focused on 3 sub-groups based on age (<13 or older) and disease susceptibility. The clinically susceptible cohort (all <13) were defined as having 3 or more reinfections in 18 months, whereas the protected cohorts were defined as having 1 or less infections in the following 18 months. Protected individuals were separated into two cohorts, <13 and older to enable age-matched comparisons with the susceptible cohort. The authors showed that the protected older cohort had broader plasma IgG reactivity and higher titers against Pf Ags, and increased functional merozoite-specific IgG. Classical monocytes, circulating Tfh and memory B cells were increased significantly in both protected cohorts, whereas memory CD4 T cells were solely increased in older protected group. These data suggest that increase classical monocytes, Tfh and memory B cells in the blood are associated with protection and support cohort divisions. The authors then went on to examine transcriptomes and TCR gene expression in naïve and effector/memory T cells from 3 protected individuals undergoing infection (not mentioned which age cohort).

Patients #1, 2 and 3 were respectively 18, 25, and 36 years old at the time of enrollment. This information has been added to the revised manuscript (Figure 5 legend and Figure S4B).

They showed that memory T cells could be clustered into 8 distinct groups, largely consisting of Th1, Th2, Tfh, Tcm and regulatory T cells. Analysis if TCR clonotypes revealed that most expanded clones belonged to what appears to be the most differentiated Th1 type of memory CD4 T cells that expressed a robust cytolytic program. These cells were implicated in malaria immunity but no transcriptional data was provided to show that these cells expanded upon malaria infection.

The authors then showed that the genes for Zeb2 and granzyme B were preferentially expressed by the expanded clones, suggesting the protein products may be useful surrogate markers for malaria-specific cells that have clonally expanded. They then showed that Zeb2+ but not granzyme B+ cells were expanded during acute infection of both younger cohorts, but not in older protected individual. The transcriptional data are unlikely to demonstrate whether memory CD4+ T cells expanded in response to malaria infection. However, this point is well taken. We have therefore set up a new functional assay, also know as "AIM" assay (Activation Induced Marker Expression) using distinct purified *Pf*-derived Ags (CSP, AMA-1, MSP-1, EBA-175 or control Ags) and tested if ZEB2+ memory CD4+ T cell activated in response to these *Pf* Ags. New data in Fig. 7 show that ZEB2+ memory CD4+ T cells, which contain the expanded clones, recognize the *Pf*-derived CSP Ag, providing formal proof of concept that ZEB2+ memory CD4+ T cells responded to *Pf* parasite Ags.

The authors propose that these findings suggest that this type of CD4 T cell population (Zeb2+) may be important for protection of younger cohorts, but don't provide any direct evidence for protection. It is a challenge to show exactly which cells or mechanisms are essential for protection in any infection. While animal models can provide much more insights into such question, human studies are often confined to correlates. However, the revised manuscript now provides new data in Fig. 8 examining the proportion of ZEB2+ memory CD4+ T cells in the PBMC of a subset of our patient cohort that either develop or fail to develop clinical protection across the 18 month longitudinal study. While the sample number remains small, the data suggest that in those individuals that gain clinical immunity, ZEB2+ memory CD4+ T cells significantly expand during a reinfection episode compared to steady state, while in those that do not, these cells failed to expand significantly.

While the study represents a nice assessment of many aspects of immunity in humans undergoing endemic infections with Plasmodium species, there are no clear links between observed phenotypes and disease or disease outcome. The first part of the paper makes some previously reported associations to support cohort divisions but the later parts of the paper are really only descriptive. There is a lot of speculation in the discussion but it is really only speculation that is loosely supported by the findings. I have highlighted some major and minor issues below.

The new set of data in revised Figs. 7 and 8 provide proof of concept that ZEB2+ memory CD4+ T cell react to the *Pf*-derived sporozoite stage CSP Ag and that these cells expand more in patient gaining clinical immunity compared to those that do not. These findings strengthen our original conclusions and are consistent with the proposed model that these cells may contribute to protection at the liver stage of the infection. In addition, we also reveal that ZEB2+ memory CD4+ T cells in age-matched clinically

immune patients express lower cell surface levels of chemotactic receptors CXCR3 and CCR6 and exhaustion markers compared to susceptible subjects.

Major issues:

Major issue 1. The authors draw the conclusion "The broader plasma IgG reactivity and titers against Pf Ags, together with the increased functional Pf-merozoite specific IgG responses in the protected compared to the susceptible participants, further verifies the clinical immunity classifications for the in depth studies of their immune cells presented below." But, these differences are generally only seen between the older protected group and the younger unprotected group, and not between the two younger age-matched groups. Thus, this softens justification for the proposed comparison and means that any significant differences could be correlated with age rather than immune status.

Example: Number of Ags recognized and amount of IgG were higher in protected older group, but not different between protected and unprotected age-matched younger groups.

Related to this issue, while the data support the conclusion as written, the following sentence is misleading as it hides the fact that there is no significant difference between age-matched protected and unprotected groups: "Approximately 50% of THP-1 cells were YOYO-1+ when co-cultured with merozoites sensitized by preincubation with clinically immune participants plasma of all ages. In comparison, only 35% of THP-1 cells stained positive for YOYO-1 after co-culture with merozoites preincubated with plasma from the susceptible group"

We have re-written this section to more clearly state that the classifications of clinical immunity are strictly based on the number of clinical infections over the 18 month longitudinal study period. Older protected patients have a greater breadth and depth of anti-malaria antibody responses, however the reviewer is correct that the differences in Ab responses between age-matched susceptible and protected patients do not reach statistical significance. This lack of difference in measured antibody responses in the age-matched group is now more accurately reported. In addition, the mechanisms of protection in young children may be due to antibody function rather than concentration or breadth, and/or differences in cell-mediated immunity.

Major issue 2. Evidence that Th1 cytolytic phenotype cells make up the majority of expanded clones in the blood is interpreted to mean that this type of effector is preferentially induced during malaria and may be important for protection (- no data on the latter though). However, this observation could relate to a preference for this phenotype of clones to largely reside in the blood, compared to other malaria-specific populations. If, for example, various subsets of memory T cells were expanded in response to malaria after infection: some that go into tissues to form tissue-resident populations, some that go into the lymph nodes and spleen (like Tcm) and some that form a vascular-resident population, then the latter may be over-represented in expanded clones in the blood. A paper by Bugget et al. suggests that this may be the case for CD8 T cells in humans (Cytolytic CD8 T cells confined to intravascular circulation. Buggert et al. Cell 183: 1946-1961, 2020). This type of outcome might mean that many different subtypes of CD4 T cells are formed in response to malaria but we preferentially only see the cytolytic CD4s expanded in the blood. This would not mean they are a critical protective subset, but simply that they are more visible when sampling blood.

It is certainly a possibility that ZEB2⁺ cytolytic memory CD4⁺ T cell clones are more visible in the blood than in other compartments and/or that other tissue-resident/expanded subsets we cannot track in this analysis, are key for protection. However, and as underlined above, the new data we added in the revised manuscript show that ZEB2⁺ memory CD4⁺ T cell clones in the blood of infected patients recognize the Pf CSP Ag (Fig. 7). Expansion of these clones also correlates with the onset of clinical immunity in a subset of patients (Fig. 8), consistent with a potentially protective role for these cells.

Major issue 3. The authors finally conclude that because a large proportion of expanded clones express Zeb2 and they find that Zeb2⁺ cells are found to be expanded in the blood of the two younger cohorts upon infection, these cells may be protective. However, their data shows no difference between protected and susceptible age-matched cohorts. This seems to argue that they are not the key population required for protection.

We report that ZEB2⁺ memory CD4⁺ T cells are expanded during infection in both susceptible and age-matched protected, yet we do not know if these cells are functionally different. The most supportive evidence comes from the newer set of data we added in the revised manuscript correlating ZEB2⁺ memory CD4⁺ T cells expansion to the acquisition of clinical immunity in a subset of patients (Fig. 8). The

fact that we did not find significantly different expansion of ZEB2⁺ memory CD4⁺ T cells in older protected patients between illness and convalescence, even though we know from our single cell analysis in Fig. 6 that these patients have expanded ZEB2⁺ clones, may be accounted for by many reasons. For instance, expansion may have been faster and already occurred in the liver of these subjects (along the same line of thoughts as point #2) and our FACS-based analysis of ZEB2⁺ cells may not have sufficient resolution to pick up smaller/more modest clonal expansion.

Minor

1. No y axis label on Figure 1E

This has been fixed.

2. Reactivity to LSA3 is implicated to be to liver-stage as wording implies it is only a liver-stage Ag but based on Malaria Cell Atlas, it appears to be expressed also during the blood stage. Might need to reword reference to this to allow reader to consider that this immunity may have been generated to blood stage Ag exposure.

We have changed wording in the revised manuscript to reflect this nuance.

3. Figure 3B. I am not sure that it is correct to do t test without correcting for multiple comparisons. This was actually corrected for multiple comparisons. We have updated the legend of the figure to reflect this.

Also, I think that Myeloid DC, pDC and NK cells need to be on a different scale to show variation as these subsets are a very small proportion of the total cells.

We have modified the scale so that variations in these subsets of cells can be seen more clearly.

4. Statement line 266 that class-switched B cells are increased in both groups is not supported by the data (Figure 1C). This figure only shows that memory B cells are increased in both protected groups. We believe this comment refers to Figure 4C. We have modified the statement to reflect data showing the increased switched B cells in protected adults compared to other groups.

5. Figure S4B, E. I think it would be clearer if the individual's data were shown on separate UMAP plots next to each other to enable comparison of cluster make up. It is not clear to me that participant 2 has cells in the Tfh cluster in S4B for example, and S4E is so full of colors it is hard to decipher.

We have now separated the individual patient's data to facilitate the comparisons (Figure S4B).

6. What age were the 3 protected individuals used for transcriptomics?

Patients #1, 2 and 3 were respectively 24.6, 18.4 and 34 years old at the time of enrollment. This information has been added to the revised manuscript.

7. I am not clear how relative expression is calculated in Figure S5B. Is there a threshold that is set as positive then the clones are assessed for how many individual cells in the clone pass this threshold? We had at least 10 identical clones for a clone to be considered "expanded". Among them, we calculated the proportion of expanded clones expressing each indicated gene. A cell was considered positive if the gene expression was detected (count RNA>0).

8. Figure 7A. What is y axis denominator? Is it white blood cells or CD4 T cells or CD4 memory T cells? Clearly outline how this relates to the proportion of Th1 cluster 1 cells you found earlier. Do these numbers match at all?

This is the proportion of ZEB2⁺ memory CD4⁺ T cells amongst all memory CD4⁺ T cells (Figs. 7A and 8A). The y axis label has now been clarified. A substantial proportion of expanded clones (that share the same TCR clonotype) are ZEB2⁺ (Fig. 6G). These proportions do match that of expanded memory CD4⁺ T cell clones among all memory CD4⁺ T cells that are shown in Fig. 6A since ~ half of expanded clones express ZEB2. This notion has also been clarified in the revised manuscript.

Why don't you have any CD4-negative T cells that are Zeb2+. Should this also come up on CD8 effectors?

The initial dot plot was an overlay of ZEB2⁺ memory CD4⁺ T cells on memory CD4⁺ T cells versus ZEB2⁺ cells, to better highlight these cells. But this representation makes other ZEB2⁺ memory T cells appear artificially faint. We have therefore used a direct gating strategy that shows ZEB2⁺ cells after gating on

Figure 1: ZEB2⁺ CD8⁺ T cells are found among activated/memory CD8⁺ T cells in the PBMC from a representative malaria-infected individual in our study cohort.

CD3⁺CD45RO⁺ cells, which is now provided in the revised paper (new Fig. 7A). As also expected, activated/memory CD8⁺ T cells also express ZEB2 (See **Figure 1** for this reviewer).

R#2

This is a really interesting and ambitious look at the combinatorial factors leading to disease resistance in *P. falciparum*. They show that patients in their study area under 13 are still largely susceptible, but can identify some individuals with fewer reinfections over the study period who have less anemia, lower parasitemia, fewer episodes of fever, and higher quality opsonizing antibodies, but not more Pf antigens recognized (for younger immunes). That is just up to Figure 2! They used a combination of multidimensional assays (CyTOF, spectral flow cytometry and single cell transcriptomics) to identify immune correlates of immunity. It is interesting (and potentially useful to note in the abstract) that there were increases in Tfh, Isotype switched B cells and classical monocytes seen in protected subjects.

We have incorporated these findings in the revised abstract.

The population that stood out of these analyses was a little-studied highly differentiated cytolytic CD4 T cell subset. While this is interesting, and MHCI recognition has been shown to kill infected *P. vivax* infected immature red blood cells, it is not clear how MHCII recognition can lead to beneficial killing of the parasite, though killing of infected Kupfer cells, or phagocytes or antigen presenting cells could have outsized effects on immunoregulation as well. It seems likely that the killing function is incidental to the highly differentiated Th1 phenotype described.

Our recent data added in the new Fig. 7 suggest that ZEB2⁺ memory CD4⁺ T cells recognize the sporozoite Pf CSP Ag rather than Pf AMA-1, MSP-1 or EBA-175 blood stage Ags. While killing of hepatocytes or Kupffer cells by these cells may be incidental, this finding is consistent with a potential role for these expanded clones during liver stage infection.

The discussion could be streamlined a bit potentially by focusing on highly differentiated Th1-like memory cells, called Teff Late, as described in Opat et al, 2018; which are protective and also can be generated by CD27- Teff Late (Opat 2015) that protect and express several Granzyme genes, as seen in the transcriptomics GEO GSE89555. Perhaps Th1-like mouse malaria papers (Zander/Butler, Wikenheiser/Stumhofer et al) can also shed some light.

We thank this reviewer for her/his insight and have discussed some of these studies/results in the revised discussion.

Alternatively, are there individual markers that are significantly differently expressed in T cells from protected participants that could be used to distinguish these children in future studies?

From our refined analysis in new Fig. 8C, D, we report that CD39⁺ ZEB2⁺ memory CD4⁺ T cells in the blood of clinically immune children express lower levels of CXCR3 and CCR6 and exhaustion markers (PD-1, LAG-3, TIGIT) than those from susceptible children. This would need further confirmation on a larger cohort with similar immune status but it is certainly a possibility.

Another idea is to look for FcR, which have been seen upregulated on T cells in malaria, NK cells with FcR can kill RBC with antibody using granzymes.

It is a great suggestion but, unfortunately, we did not stain for FcR expression in our mix when we conducted the CyTOF experiments several years ago. This would require substantial time and investment to incorporate this marker in the CyTOF panel or even incorporate NK and FcR in our Cytek panel. We also do not have more samples for already analyzed patients during illness. We nevertheless checked in the differentially expressed genes corresponding to this cluster #1 of memory CD4⁺ T cells and found evidence for FcR-like 6 and FcR-like 3 encoding gene expression (Table S5), consistent with this reviewer's suggestion.

The paper does not seem to come full circle in identifying T cell features that are different between protected and susceptible children, but the groundwork has been nicely laid for future studies to determine potentially protective attributes discovered here. Were expanded memory T cell subsets only seen in protected participants, but not susceptible?

The single cell sequencing analysis was done on protected adult patients. We next used ZEB2 as a surrogate marker to track expanded clones and compare ZEB2⁺ memory CD4⁺ T cells expansion in young protected versus susceptible patients between illness and convalescence (30 days later). While we found these cells expanded in both groups during illness compared to convalescence, we did not measure differences between these groups. We also did not find expansion of ZEB2⁺ memory CD4⁺ T cells in protected adults, perhaps as they do not expand as much (though clones with shared TCR clonotypes were discovered in the protected adults) or the kinetic of expansion and contraction was

faster. During the revision of this work, we further analyzed a small subset of young age-matched patients who either develop clinical immunity or not over the 18-month timeline of our study (new Fig. 8B). We report that greater ZEB2⁺ memory CD4⁺ T cell expansion correlated with the acquisition of clinical immunity in these subjects, consistent with the hypothesis that these cells may contribute to clinical immunity. We also established that ZEB2⁺ memory CD4⁺ T cells recognize the *Pf*-derived CSP Ag, strengthening our findings.

Are there inhibitory co-stimulatory molecules that are significantly changed in protected but not susceptible participants (Zeb2 and GzmB? Or CX3CR1/CXCR3, CXCR5, CCR6)? If so, list in abstract. When refining our the cell-surface phenotypes of ZEB2⁺ memory CD4⁺ T cells in young protected versus susceptible children reported in the new Fig. 8C, D, we noted that CD39⁺ ZEB2⁺ memory CD4⁺ T cells in the blood of clinically immune children express lower levels of the “inflammation homing” chemokine receptors CXCR3 and CCR6 and of the exhaustion markers (PD-1, LAG-3, TIGIT) than those from susceptible children. This may suggest that while access of these cells to infected tissues may be more restricted, their functional capacity may be superior. We have updated our interpretation of these data, and modified the abstract of the revised manuscript accordingly.

Minor suggestions

-Show controls in S2C in Figure 2C

We tried to add S2C in 2C but felt it overloaded the figure. Since this really is control information, we decided to leave it in S2C.

-Perhaps its confused by the lack of specificity in the T cell analysis, but GATA3⁺ Th2 cells and Tcf7⁺ Tcm seem unexpected in malaria. Do Th2 correlate with worm exposure (if you have this clinical info)? Do these cells also express IL-4, 5 or 13, upregulated STAT6? ST2? If not, perhaps identifying the population as GATA3⁺?

The single cell transcriptomic analysis was done on all activated/memory-phenotype CD4⁺ T cells, during an episode of *Pf* malaria. While it is expected that a fraction of CD4⁺ T cells will respond against the parasite, not all of them will react to *Pf* malaria Ags. Rather, we expect that these cells will include various known subsets of Ag-experienced T cells that may or may not be related to malaria. Through dimensional reduction analysis, we defined 8 distinct subsets based on single cell gene expression differences, and the differential expression of specific TF and genes encoding for known cell surface markers helped us to relate the various clusters to known/major CD4⁺ T helper cell subsets. These included subsets with T_{CM} and T_{H2} (GATA-3) gene expression signatures, as listed in Fig. 5C. While we do not know the worm exposure status of these patients, it is possible that these T_{H2}/GATA-3⁺ memory CD4⁺ T cells are recognizing worm-derived antigenic peptides. In addition to GATA-3 transcripts, we also detected IL-13 transcripts but could not confirm at the protein level as a result of limited material. We have carefully reworded this section of the result to reflect some of these limitations and address this reviewer's concerns. We also note that the new data in Fig. 7 formally establish that ZEB2⁺ memory CD4⁺ Th1 cells, which contain a significant proportion of expanded clones, recognize the *Pf*-derived CSP Ag, supporting further our focus on these cells.

Not sure we can call Tcf7 a master regulator of Tcm yet.

We have tweaked the revised version to avoid any overstatement.

Surprising that quiescent Tcm would be a large subset in malaria too, particularly in a region of repeated exposure. What does the metabolism transcriptome suggest? The cell cycle genes?

Memory CD4⁺ T cells often include a substantial proportion of T_{CM} phenotype cells as also reported by many studies. It is likely that most of these cells do not recognize malaria Ags. We also did not find any cell cycle associated genes, however, we noted differential expression of the gene encoding for the PAS Domain Containing Serine/Threonine Kinase that regulates glucose and energy metabolism.

--results section for costimulatory and inhibitory molecules (lines 394-401) is not clear, do young protected participants have more or less inhibitory or costimulatory molecules on their expanded clones or total Tmem?

We have used ZEB2 as a surrogate marker to track expanded *Pf*-specific memory CD4⁺ T cell clones. Our refined analysis reveals that ZEB2⁺ memory CD4⁺ T cells in the blood of clinically immune children express lower levels of the chemokine receptors CXCR3 and CCR6 and of the exhaustion markers (PD-1, LAG-3, TIGIT) than those from susceptible children. This section has been re-written and incorporate the new figs 7 and 8.

R#3

Furtado et al have performed a comprehensive longitudinal study of participants living in a malaria endemic area of Malawi. Participants were classified as either susceptible or protected from *Plasmodium falciparum* (Pf) infection based on the number of recurrent clinical infections over the 18-month study period. Their findings demonstrate age related differences in susceptibility to *Plasmodium falciparum* infection and clinical immunity was also dependent on these age differences. They performed detailed analysis on plasma and peripheral blood mononuclear cells (PBMCs). Compared to susceptible participants, they found higher plasma IgG titres and broader IgG to Pf antigens in protected participants as well as greater antibody dependent Pf opsinisation. High dimensional flow cytometry was performed on cryopreserved PBMCs for immunophenotyping and single cell transcriptomics was performed on 3 protected participants. The study shows some interesting results, in particular an expansion of memory CD4 T cell clones with cytolytic characteristics.

Specific comments are found below:

Results

- Line 140: the authors state they enrolled equal numbers of children, adolescents and adult participants but the data in figure S1A is confusing as it shows a table with the participant age break down as 1-5 years, 6-12 years and 13-50 years. Generally, the definition for a child would be <12 years old, adolescent 13-18 years old and adult >18 years old. Can the authors explain the reason for the breakdown of age brackets in the table shown in S1A?

Our goal was to capture sufficient numbers of subjects that were clinically immune and clinically protected during malaria infection. As this region of Malawi is highly endemic, we expected that clinical protection was acquired at a young age, and thus we oversampled young children and adolescents (n=35 in each group): young children (1-5 yrs) and adolescents (6-12 yrs) along with adults (13-50 yrs). As the age of clinical immunity is dependent on local factors (transmission intensity, host genetics, malaria interventions), we were then able to examine our own dataset and categorize clinically susceptible or clinically immune participants based on the number of clinical reinfections. The revised manuscript describes more carefully the rationale for over sampling younger children.

- It becomes apparent in the next section of the manuscript (line 153) that new definitions need to be set for each malaria endemic region and this then sets new age brackets and 3 populations that are further studied in detail. Perhaps for clarity it would be better to include a table of the study characteristics (as those listed in S1A) of the 3 groups – Susceptible <13yrs, Protected <13yrs and Protected >13years.

We agree with the reviewer that this was confusing. We have added a table (Fig. S1D) that reports the demographics and variables characteristics of the clinically immunity study groups definitions, for which all the immune studies are based on as: <13yr, Susceptible, < 13 yr Protected, Protected ≥13yr)

- Line 213: What is the cut-off for the YOYO-1 fluorescent dye/assay? Has this been defined?

We provide in **Figure 2** for this reviewer representative FACS dot plots with i) only THP1 cells and no labelled RBC and ii) THP-1 cells incubated with *Pf*-infected merozoites only (no serum), that we used to set up the positive versus negative gates shown in Figure 2C.

- Line 234: Table S4 has been labelled Table S5

This has been corrected.

- Line 236: CD62L expression is generally best detect on fresh blood and difficult to detect on frozen PBMCs, it is unclear from the gating strategies presented whether CD62L was even used for gating – please clarify

Figure 2: ZEB2⁺ CD8⁺ T cells are found among activated/memory CD8⁺ T cells in the PBMC from a representative malaria-infected individual in our study cohort.

We did not use CD62L for our gating for this exact reason, although CD62L expression on naïve T cells was detected.

- Line 245 – 253: the authors discuss significantly higher frequencies, however no values are included, in my opinion the authors should report the medians and p-values in the text.

We have added the proportion of the various cell subsets in the text ; p-values are also indicated on the figures and legends when appropriate.

- Line 256: It must be noted that the TFH found in the periphery are not bona fide TFH cells but circulating TFH - also it is unclear if these have been gated on total CD4 T cells or memory CD4 T cells, this could lead to a difference in results between groups – please clarify.

We have modified T_{FH} to circulating T_{FH} (cT_{FH}) in the revised text. This was gated on all CD4⁺ T cells, which is now clarified in the revised text. When we gate on memory CD4⁺ T cells, the results are similar since most of cT_{FH} cells are Ag experienced (See **Figure 3** for this reviewer).

- Gating of Treg cells is also unclear if based on total CD4 or memory CD4 – please clarify. Also gating of Treg cells is more reliable if CD25 and CD127 are included see Sedikki et al

Figure 3: Increased blood proportion of cT_{FH} (all and activated/memory or Mem) in protected versus susceptible subjects during a *Pf* malaria episode.

<https://www.ncbi.nlm.nih.gov/pmc/articles/PMC2118333/pdf/jem2031693.pdf>

This was gated on all CD4⁺ T cells. To our knowledge, FOXP3 which is the lineage specifying transcription factor of naturally occurring and some induced T_{reg} cells, appears as the most reliable and precise marker to track T_{reg} cells. The paper cited by this reviewer uses the combination of cell-surface expression of CD25 and CD127, and CD45RA and CD45RO, to identify T_{reg} cells as CD25⁺CD127^{lo} and to discriminate them from activated conventional T cells (CD25⁺CD127⁺CD45RO⁺). The authors propose this cell surface gating strategy as an alternative to staining for Foxp3 intracellular expression, but not as a superior approach. In this CyTOF experiment, we did not stain for CD25 and CD127 expression, therefore we could not make such comparison.

- Line 287: Assume this is the correct Table S5 as per uploaded version. This is correct.

- Line 316: Does the TFH cluster express PD-1 and CXCR5?

Our analysis did not reveal transcripts encoding for these genes. Differential expression of transcripts encoding for TCF7, IL7R, BAFF and CCR6 was consistent with a circulating T_{FH} cell signature.

- Line 368: Have the authors checked to see if these clones are specific for Pf?

The revised manuscript now includes new functional assays (Activation Induced Marker Expression or “AIM” assay) that we set up using various purified/vaccine grade malaria antigens (CSP, AMA-1, MSP-1, EBA-175). We tested memory CD4⁺ T cell activation in response to these *Pf* Ags. We provide new data in revised Fig. 7 showing that ZEB2⁺ memory CD4⁺ T cells, which contain the expanded clones, are enriched in *Pf*-derived CSP reactive T cells, strongly suggesting that these expanded clones responded to *Pf* sporozoite-derived parasite Ags.

- Line 385: Same comment as above for CD62L

As also mentioned above, CD62L expression on naïve T cells was detected.

- Line 387: Are the authors suggesting the Zeb2+ CD4 T cells that upregulate CD39 are specific for Pf? In combination with CD25 and CD134, CD39+ CD4 T cells have been shown to be antigen-specific Tregs (<https://pubmed.ncbi.nlm.nih.gov/24752698/>).

This was indeed our suggestion. The additional data in the revised paper show that ZEB2⁺ memory CD4⁺ T cells react to *Pf* CSP-1 Ag (new Fig. 7) and express higher CD39 (new Fig. 8C). Upon Ag-mediated activation *in vitro* (AIM assay), they underwent clear upregulation of CD25 and OX40/CD134.

Also, our single cell transcriptomic analysis did not reveal co-expression of ZEB2 and FOXP3 encoding transcripts and we did not find robust evidence for CD25 expression on ZEB2⁺ memory CD4⁺ T cells.

- Line 394-401: No reference to the figures.

We have re-written this section to clarify our findings and made sure to include reference to the data.

- Line 405: There is no indication that these cells are antigen specific, would be a nice addition to the current data if possible.

See our answer above (line 368 comment).

- Line 475: would be great to show that these cells actually have cytotoxic potential if possible, perhaps in a killing assay?

We added evidence for *Pf*-specificity in the new Fig. 7. While we agree with this reviewer, running such assays would require much more cells than what we currently have unfortunately.

- Line 601, 618, 624: Assume Table S5 is actually Table S4 (as per upload)

This is correct, we apologize for this mistake.

Methods:

- Unclear if experiments were performed on PBMCs from different time points, please clarify

Figures

All experiments analyzing PBMC were done at enrollment during the first malaria episode of the longitudinal collection, except for some and the newer data presented in Figs. 7 and 8. In new Fig. 7 (AIM assay), we used PBMC from patients of the longitudinal analysis during a malaria episode. In new Fig. 8, A/B panels analyze PBMC from enrollment and 1 month convalescence. C uses PBMC from recurrent illnesses across the 18 month study. We have added this information and all specifics for each sub-group of patients used across the study when it was missing.

- Figure 1C: interesting there seems to be 2 groups in the susceptible population with regards to temperature – can the authors comment?

Interesting comment, this may simply be due to the cyclical nature of fever in clinical malaria and at the time of enrollment some participants were afebrile. For this study we did not require fever at the time of enrollment. Criteria were clinical symptoms consistent with malaria and microscopy confirmed parasitemia.

- Figure 1E: the label for the left y-axis is missing

This has been added.

- Figure 2A: x-axis missing

The legend is on the top.

- Figure 4B: Authors use CD45RA to gate for naïve CD8 T cells, can the authors be sure these are not CD8 revertants or Temra?

Unfortunately, we do not have this information, as we did not include CD28 or CCR7 in our panel.

- Figure 7A: x-axis missing from graph on right

This has been added.

- Figure 7B: bit confusing what the message is here are the Zeb2+ protected data from all ages?

We have refined our analysis and a revised figure with some of these data are now in Figure 8C, comparing aged-matched susceptible to protected subjects.

- Figure 7C: is the key missing for susceptible and protected?

See prior comment.

Figure S6B: x-axis missing from graph on right

We have removed these data from the revised manuscript as it diverted from the main message.

R#4

This assessment is only in regard to the systems biology component of this work.

Results

The bioinformatics analysis is sound, although the methods section lack key details.

We have completed the methods section of the revised manuscript to include all missing information, including details on the pseudotime analysis.

The scRNAseq +TCR was performed only on 3 protected individual, which somehow contrast with the main narrative of the manuscript. It would be more sound to consider more scRNAseq data by splitting <13 and >13 , or compare susceptible versus protected. The bioinformatic analysis should likely reveal clear signatures, including in the cytotoxic CD4

We agree with the reviewer that 3 samples will not fully recapitulate the diversity observed in the expanded population, this is why we decided to confirm our findings in an expanded cohort with a different technology, providing us with both biological and technical validations. We conducted single cell RNA-seq to identify populations of interest in protected individuals. Then, we confirmed our findings on a greater number of samples measuring protein expression levels at the single cell level using high dimensional spectral flow cytometry. The use of ZEB2 as a proxy to identify expanded clones was largely confirmed in our PBMC analysis during illness of young patients and in patients developing clinical immunity over the length of our

longitudinal study. We also established that ZEB2⁺ memory CD4⁺ T cells recognize the *Pf*-derived CSP Ag, further supporting our model. We also have to keep in mind that single-cell RNAseq is highly impacted by the quality of the samples as we cannot proceed sample with a viability lower than 70% while, this limitation can usually be overcome in Flow cytometry analysis enabling to interrogate more of the precious human samples.

How was the comparison between clonally expanded and non expanded cells performed? What test?

Clonally expanded cells were defined based on the TCR analysis. We defined clonally expanded cells as cell associated to a clonotype with a frequency >10. This threshold corresponds to the inflexion of the distribution tail for clonotype frequency.

Zeb2 was identified but was this significant?

ZEB2 was identified using differentially expression analysis comparing expanded versus non-expanded memory CD4⁺ T cell function "Find markers" in Seurat. The difference was significant with P-value $p=5.321 \times 10^{-22}$, Table S5). We have added this information and details on the analysis in the Materials and Methods of the revised manuscript.

Could naive T cells be used as root? It would be important to see if the same clone can be shared with the naive cells

We could have used naïve as root for the pseudotime analysis. However, as naïve cells cluster with T_{CM} cells (revised Figure S4D), we decided to perform the pseudotime analysis only on expanded memory cells to limit the influence of batch integration. Considering the clone information, the pseudotime analysis allows us to place each cell/cluster along the differentiation process but not to track clonotypes per se. Since the current analysis does not capture all clonotypes, if we do not find a clonotype among the naïve cells, this will not mean it is not there.

The method section should include more details.

- scRNAseq bioinformatics method is not fully explained.

The revised manuscript includes much more details on the scRNA-seq methods of analysis.

How is the trajectory analysis performed?

We have further detailed the Methods section regarding Pseudotime analysis. Briefly, differentiation trajectory analyses were conducted with Monocle. Monocle's *orderCells* function was used to arrange cells along a pseudo-time axis to indicate their position in a differentiation continuum. Monocle generates for each cell a pseudotime value in respect to predefined cells of origins (roots). Here T_{CM} cells were used as roots since they are well-known to keep the highest differentiation potentials. This pseudotime can then be interpreted as how far in the differentiation process each cell is from the root cells.

What is the rationale behind trajectory analysis?

The rationale behind the trajectory analysis is to better understand the relationship between each cluster in regard to differentiation and to understand the sequence of regulatory changes that occur as cells transition from one state to the next.

How is the route chosen?

Pseudotime is an abstract unit of progress corresponding to the distance between a cell and the start of the trajectory, measured along the shortest path. The trajectory's total length is defined in terms of the total amount of transcriptional change that a cell undergoes as it moves from the starting state to the end state. Monocle assumes that the trajectory has a tree structure, with one end of it the "root", and the others the "leaves". Monocle's job is to fit the best tree it can to the data. This task is called *manifold learning*. A cell at the beginning of the biological process starts at the root and progresses along the trunk until it reaches the first branch, if there is one. That cell must then choose a path and moves further and further along the tree until it reaches a leaf. A cell's pseudotime value is the distance it would have to travel to get back to the root.

How is the clonal analysis performed?

What statistical tests were performed and what cut off were used.

We have now detailed the clonotype analysis in the Methods section of the revised manuscript. We use clonotype frequency distribution to define clonally expanded cells. The threshold used based on the distribution tail was a clonotype frequency greater than 10. Differential expression analysis between expanded versus non expanded cells was performed using likelihood-ratio test for single-cell gene expression. P-values for reported genes were <0.01.

- Batch effect and other QC steps, this reviewer did not find any details on these QC steps of the scRNAseq. Were data batch corrected? Were cells excluded and with what details?

We added in QC steps we conducted in the revised methods section for the Single-cell analysis part. Dataset were integrated within each condition, naïve and memory representing a total of 6,664 and 7,934 cells respectively using the Seurat (v3) Canonical Correlation Analysis (CCA) and graph-based integration tool on the 3,000 most expressed genes across datasets to correct for batch effect. The 30 first dimensions of the PCA of the batch effect corrected matrix was used to generate the Shared Nearest-neighbor (SNN) graph and the UMAP. Filtering for low quality cells according to the distribution of number of RNA (between 200 and 2500), and percentage of mitochondrial RNA ($<0,05$) was performed. Then, gene expression matrix was corrected for cellular sequencing depth and mitochondrial percentage using linear modeling as implemented in Seurat ScaleData function.

REVIEWER COMMENTS

Reviewer #3 (Remarks to the Author):

The authors have improved their original manuscript with new studies and data including detection of antigen-specific CD4 T cells. The revised manuscript mostly reads well and is easy to follow. Thank you to the authors for addressing my original comments.

There are a few new points that need addressing or require further clarification, please see below:

- Figure 1C: there is no label on the x-axis
- Lines 171-173: "To determine if the greater number of clinical infections in younger participants compared to protected individuals was not due to greater exposure to malaria rather than immunity", the phrasing of this sentence is a bit confusing, the authors should consider rephrasing for clarity.
- The new table in Figure S1D does not have the age label "<13 yr" next to "Susceptible".
- Lines 188-190: sentence not well written, perhaps " This enabled us to conduct in depth immunophenotyping of their peripheral blood mononuclear cells (PBMCs), analysis of their plasma Ab and single cell transcriptomic analysis to investigate host immunity..."
- Line 248: In their response to reviewers, the authors indicate that they did not use CD62L for their gating, so it is unclear why they include CD62L as one of the markers to discriminate effector and memory lymphocytes.
- Line 274: The authors use the term Circ-TFH for circulating TFH, however the convention is cTFH – please consider changing.
- Line 400: The authors use an AIM assay, please include reference to methods papers such as Zaunders et al 2009 PMID: 19635903 DOI: 10.4049/jimmunol.0803548
- Figure 7B: The authors do not show the healthy control response to CSP, can the authors please comment what the healthy control response to this antigen is?
- Line 368 (original version of manuscript): Have the authors checked to see if these clones are specific for Pf? The authors responded to this query by performing a functional T cell assay, in particular an AIM assay. However, they do not show that the specific TCR clones detected in the previous section are identified in the CD25+CD134+(OX40) population. This would have required sorting and TCR sequencing of the Pf-specific CD4 T cells. Alternatively the authors could have looked at CD4 T cell databases to match Pf-specific TCRs to the ones they describe, if such a database existed.
- Figure 8A has no x-axis

Reviewer #4 (Remarks to the Author):

The authors have addressed my comments.
The methodology has been more detailed.

Of note, some of the method sections still lack important details such as parameter values used, version of software utilised.

also, Monocle in bioconductor is the obsolete version 2.

There is a version 3. Also, the authors may want to consider a simpler application of pseudotime trajectory analysis, such as slingshot or other similar. This because the structure of their data is very clearly showing differentiation towards the clonally expanded population in the cluster south of the UMAP.

Reviewer #5 (Remarks to the Author):

The authors have tried to address the comments of reviewers 1 and 2. I agree with reviewer 1 who had 3 major issues and I don't think it is possible to easily answer these questions in field study data such as this shown here. There is not really a correlation of protection shown and the functional capacity of cytolytic CD4 T cells is not demonstrated so highly speculative if indeed these cells do play a role in limiting successful maturation in hepatocytes. The authors have added an AIM assay to help demonstrate reactivity to Plasmodium antigens and show CSP reactivity. I think the title is generally accurate that the data in this paper shows expansion in Plasmodium infection. I would soften a few of the sentiments related to protection – I really don't think this data shows anything other than the effects of inflammation on the appearance of cytolytic CD4 T cells since clinical illness is associated with higher inflammatory responses. It is not clear to me why this would differentially affect CSP-reactive CD4 T cells and not CD4 T cells that react to blood stage antigens- do the authors hypothesize some kind of liver inflammation in response to infected blood stages accumulating in the liver is influencing this phenotype? Figure 8 is a key figure but figure 8B is confusing – I have spent some time trying to understand what samples these are but from the description provided it really is not clear how these individuals were selected and the numbers of ZEB2+ memory CD4 T cells are a bit out of whack with the numbers in all groups shown in 8A. How exactly are they defined as gaining clinical immunity (or not?). The table in Fig. S6 appears that the phrase gaining clinical immunity means older (av age 6.2 years as opposed to 2.7 years). Is it primarily age? Older individuals have less clinical episodes. Is this what the authors are using as a correlate of clinical immunity. This just needs more explanation for clarity. Despite the issues with the conclusions that can be made from the data included in this manuscript, the considerable difficulty in obtaining the data in this manuscript should be taken into consideration. It is difficult to do studies in field studies such as this other than those that are correlative. I think the manuscript does add to the literature. Documenting the appearance of these cells in human Plasmodium infections supports the rationale of taking this observation into mouse studies to determine the role that these cells may play in protection against infection and development of Plasmodium in the liver.

Point by Point Reply to Reviewer's Comments:

REVIEWER COMMENTS

Reviewer #3 (Remarks to the Author):

The authors have improved their original manuscript with new studies and data including detection of antigen-specific CD4 T cells. The revised manuscript mostly reads well and is easy to follow. Thank you to the authors for addressing my original comments.

We thank this reviewer for the positive comments.

There are a few new points that need addressing or require further clarification, please see below:

- Figure 1C: there is no label on the x-axis

The legend is specified on the top of the bar graphs, mostly to avoid multiple repeats of the same label.

- Lines 171-173: "To determine if the greater number of clinical infections in younger participants compared to protected individuals was not due to greater exposure to malaria rather than immunity", the phrasing of this sentence is a bit confusing, the authors should consider rephrasing for clarity.

We have re-written this sentence to better convey our point.

- The new table in Figure S1D does not have the age label "<13 yr" next to "Susceptible".

This has now been added.

- Lines 188-190: sentence not well written, perhaps " This enabled us to conduct in depth immunophenotyping of their peripheral blood mononuclear cells (PBMCs), analysis of their plasma Ab and single cell transcriptomic analysis to investigate host immunity..."

We have rewritten this sentence to clarify our point.

- Line 248: In their response to reviewers, the authors indicate that they did not use CD62L for their gating, so it is unclear why they include CD62L as one of the markers to discriminate effector and memory lymphocytes.

We apologize, CD62L should have been removed from this sentence as we have not used it to discriminate effector versus memory cells (also shown from our detailed gating strategy depicted in Figure S3C.

- Line 274: The authors use the term Circ-TFH for circulating TFH, however the convention is cTFH – please consider changing.

We have made this change in the text and figure.

- Line 400: The authors use an AIM assay, please include reference to methods papers such as Zaunders et al 2009 PMID: 19635903 DOI: 10.4049/jimmunol.0803548

We have now included this original reference, thank you for pointing this out.

- Figure 7B: The authors do not show the healthy control response to CSP, can the authors please comment what the healthy control response to this antigen is?

Healthy control response to CSP is similar to that of all Pf-derived Ag shown in Fig. 7B. We have only run two of these controls, and since the response was similar to all Pf Ag, we did not test more of these conditions. We provide a staining example for this reviewer below:

- Line 368 (original version of manuscript): Have the authors checked to see if these clones are specific for Pf? The authors responded to this query by performing a functional T cell assay, in particular an AIM assay. However, they do not show that the specific TCR clones detected in the previous section are identified in the CD25⁺CD134⁺(OX40) population. This would have required sorting and TCR sequencing of the Pf-specific CD4 T cells. Alternatively the authors could have looked at CD4 T cell databases to match Pf-specific TCRs to the ones they describe, if such a database existed.

As rightly pointed out by this reviewer, we have not directly demonstrated that the expanded clones we detect by single cell TCR sequencing analysis are CSP-specific, which would require the sorting of these cells using CSP tetramers and TCR sequencing. This would also require to know which CSP peptides are recognized and which HLA-II molecules are presenting the CSP stimulating epitopes in our Malawian cohort. These are very interesting questions and clearly the next steps of this work which will likely necessitate several more years of work. As for the other suggestion, we are not aware of such database of Pf-specific CD4⁺ T cells, we believe that our study is the first of that kind in human malaria patients.

- Figure 8A has no x-axis

We have added a x-axis. Legend is shown on the top right corner of the graph.

Reviewer #4 (Remarks to the Author):

The authors have addressed my comments.

The methodology has been more detailed.

Of note, some of the method sections still lacks important details such as parameter values used, version of software utilised.

We have added the few missing information related to the single cell analyses.

also, Monocle in bioconductor is the obsolete version 2.

There is a version 3.

We did use the Monocle version 3, we apologize for this reporting error.

Also, the authors may want to consider a simpler application of pseudotime trajectory analysis, such as slingshot or other similar. This because The structure of their data is very clearly showing differentiation towards the clonally expanded population in the cluster south of the UMAP.

We thank this reviewer for the suggestion. There is no significant differences between Slingshot or Monocle3 in term of performance. Monocle3 and Slingshot are among the top 4 best methods and give very comparable results with Monocle3 reported to have a slight edge over Slingshot (*BMC Bioinformatics* 24, 55 (2023). <https://doi.org/10.1186/s12859-023-05179-2>). Regarding the implementation of Monocle3, we did not encounter any major difficulties, we believe that the reviewer

was referring to Monocle 2 vs Slingshot (as we incorrectly referred to this version in the manuscript), but the Trapnell's team has made significant improvement in packaging version 3 versus version 2.

Reviewer #5 (Remarks to the Author):

The authors have tried to address the comments of reviewers 1 and 2. I agree with reviewer 1 who had 3 major issues and I don't think it is possible to easily answer these questions in field study data such as this shown here. There is not really a correlation of protection shown and the functional capacity of cytolytic CD4 T cells is not demonstrated so highly speculative if indeed these cells do play a role in limiting successful maturation in hepatocytes. The authors have added an AIM assay to help demonstrate reactivity to Plasmodium antigens and show CSP reactivity. I think the title is generally accurate that the data in this paper shows expansion in Plasmodium infection.

We thank this reviewer for his generally positive assessment. Finding whether ZEB2⁺ memory CD4⁺ T cells contribute to protection against malaria infection is a key question that we ought to pursue but that is clearly beyond the scope of the current 7+ year study.

I would soften a few of the sentiments related to protection – I really don't think this data shows anything other than the effects of inflammation on the appearance of cytolytic CD4 T cells since clinical illness is associated with higher inflammatory responses. It is not clear to me why this would differentially affect CSP-reactive CD4 T cells and not CD4 T cells that react to blood stage antigens- do the authors hypothesize some kind of liver inflammation in response to infected blood stages accumulating in the liver is influencing this phenotype?

We did not find evidence for higher ZEB2⁺ memory CD4⁺ T cell reactivity against the pool of four Pf Ags including CSP than CSP only (Fig. 7B). While reactivity to CSP is likely dominant among the Ags we tested, it is possible that a smaller proportion of ZEB2⁺ memory CD4⁺ T cells react against the three blood stage Ags in the mix or against other blood stage Ags that we could not assess. Our thinking is that CSP, a pre-erythrocytic Ag, is expressed at early stage of infection thus drives a robust and rapid CSP-specific CD4⁺ T cell response that is dominant. If this response contributes to protection, this may blunt other parasite-specific T cell responses. This interpretation which we discuss, remains highly speculative as pointed out by this reviewer, and much more work will be required to validate such model. We have tried to clarify our thinking and carefully soften the take-home message accordingly.

Figure 8 is a key figure but figure 8B is confusing – I have spent some time trying to understand what samples these are but from the description provided it really is not clear how these individuals were selected and the numbers of ZEB2⁺ memory CD4 T cells are a bit out of whack with the numbers in all groups shown in 8A. How exactly are they defined as gaining clinical immunity (or not?). The table in Fig. S6 appears that the phrase gaining clinical immunity means older (av age 6.2 years as opposed to 2.7 years). Is it primarily age? Older individuals have less clinical episodes. Is this what the authors are using as a correlate of clinical immunity. This just needs more explanation for clarity.

This point is well taken, and we now more clearly describe how these samples were selected, the definition of “gaining clinical immunity”, in the results and in methods sections. To further explore the association between ZEB2⁺ memory CD4⁺ T cells and clinical immunity, we first identified subjects from the susceptible group who were gaining (or predicted to gain) immunity during the 18 month study. We selected subjects who had at least 2 episodes of clinical malaria and using a mixed effects model, we identified subjects who appeared to develop clinical immunity and others who remained clinically susceptible over 18 months and had available PMBCs for immunophenotyping. Figure 8B shows that both individuals gaining protection and not gaining protection have clinical malaria during the early months of the study (infection associated with a temperature of ≥ 37.5 °C with a *P. falciparum* parasitemia density $\geq 2,500$ parasites/ μ L), whereas later during the study the subjects gaining protection

are infected without fever and are only PCR positive for malaria in contrast to the subjects that are not gaining protection who continue to have infection associated with fever and higher parasitemia. We were not surprised that the age window of gaining immunity is older than those who do not yet gain immunity. We agree that age is playing a role in the development of clinical immunity, therefore we propose that the appearance of these cells may be playing a role. We were able to identify five and four such subjects that matched either criteria. We analyzed the proportions of ZEB2⁺ memory CD4⁺ T cells in their PBMC during the enrollment episode of clinical malaria and during a subsequent clinical malaria episode that occurred later in the study. The results revealed increased expansion of ZEB2⁺ memory CD4⁺ T cells in individuals who gained clinical immunity compared to those who did not. We also note that, except for one subject in each group, relative frequencies of ZEB2⁺ memory CD4⁺ T cells were not very different than our findings in Figure 8A.

Despite the issues with the conclusions that can be made from the data included in this manuscript, the considerable difficulty in obtaining the data in this manuscript should be taken into consideration. It is difficult to do studies in field studies such as this other than those that are correlative. I think the manuscript does add to the literature. Documenting the appearance of these cells in human Plasmodium infections supports the rationale of taking this observation into mouse studies to determine the role that these cells may play in protection against infection and development of Plasmodium in the liver.

We thank this reviewer for this supportive final comment.

REVIEWERS' COMMENTS

Reviewer #3 (Remarks to the Author):

Thank you to the authors for addressing my comments on their revised manuscript. I am satisfied that the authors have addressed my concerns in their revisions. I congratulate the authors on this interesting and important study and I am happy with the latest version of their manuscript.

Reviewer #4 (Remarks to the Author):

The authors have addressed all my remaining comments and I am happy with the current version of the manuscript.

Reviewer #5 (Remarks to the Author):

The authors have addressed my comments and I am supportive of publishing this manuscript as is.

Point by Point Reply to Reviewer's Comments:

REVIEWERS' COMMENTS

Reviewer #3 (Remarks to the Author):

Thank you to the authors for addressing my comments on their revised manuscript. I am satisfied that the authors have addressed my concerns in their revisions. I congratulate the authors on this interesting and important study and I am happy with the latest version of their manuscript.

Reviewer #4 (Remarks to the Author):

The authors have addressed all my remaining comments and I am happy with the current version of the manuscript.

Reviewer #5 (Remarks to the Author):

The authors have addressed my comments and I am supportive of publishing this manuscript as is.

We thank all reviewers for their supportive comments.